# The sensitivity of the ENSO to volcanic aerosol spatial distribution in the MPI Grand Ensemble

Benjamin Ward[1], Francesco S.R. Pausata[1], Nicola Maher[2]

[1] Department of Earth and Atmospheric Sciences, University of Quebec in Montreal, Montreal, Canada

[2] Max-Planck-Institute for Meteorology, Hamburg, Germany

*Correspondence to*: Benjamin Ward (ward_soucy.benjamin@courrier.uqam.ca)

**Abstract.** Using the Max Planck Institute Grand Ensemble (MPI-GE) with 200 members for the historical simulation (1850-2005), we investigate the impact of the spatial distribution of volcanic aerosols on the ENSO response. In particular, we select 3 eruptions (El Chichón, Agung and Pinatubo) in which the aerosol is respectively confined to the Northern Hemisphere, the Southern Hemisphere or equally distributed across the equator. Our results show that relative ENSO anomalies start at the end of the year of the eruption and peak the following one. Especially, we found that when the aerosol is located in the Northern Hemisphere or is symmetrically distributed, relative El Niño-like anomalies develop while aerosol distribution confined to the Southern Hemisphere leads to a relative La Niña-like anomaly. Our results point to the volcanically induced displacement of the ITCZ as a key mechanism that drives the ENSO response, while suggesting that the other mechanisms (the ocean dynamical thermostat, the cooling of tropical northern Africa or of the Maritime Continent) commonly invoked to explain the post-eruption ENSO response may be less important in our model.

## 1 Introduction

Aerosol particles from volcanic eruptions are one of the most important non-anthropogenic radiative forcing that have influenced the climate system in the past centuries (Robock, 2000). Oxidised, sulfur gases (mainly in form of $SO_2$) injected into the stratosphere by large Plinian eruptions form sulfate aerosols ($H_2SO_4$) (Pinto et al. , 1989; Pollack et al., 1976) that have a time residence of 1-3 years (Barnes & Hofmann, 1997; Robock & Yuhe Liu, 1994). These particles both scatter and absorb incoming solar radiation as well as part of the outgoing longwave radiation (Stenchikov et al., 1998; Timmreck, 2012). For intense and sulfur-rich volcanic eruptions, the net effect is a general cooling of the surface and a warming in the stratosphere where the aerosols tend to reside longer (Harshvardhan, 1979; Rampino & Self, 1984). The maximum global cooling seen in modeling studies is generally reached within 6-8 months following the eruption peak in optical depth before returning to normal values after about 3 to 4 years (Thompson et al., 2009). These rapid modifications in temperature may induce dynamical changes in the atmosphere and in the ocean including a strengthening of the polar vortex (e.g., Christiansen, 2008; Driscoll et al. , 2012; Kodera, 1994; Stenchikov et al., 2006), a weakening in the African and Indian Monsoon (e.g. Iles et al., 2013; Man et al., 2014;Paik et al., 2020; Trenberth & Dai, 2007; Zambri & Robock, 2016) as well as forced changes on the El Niño-Southern Oscillation (ENSO) (e.g., Emile-Geay et al., 2008; McGregor & Timmermann, 2011; Pausata et al., 2015; Wang et al., 2018).

Paleoclimate archives and observations from the past centuries suggested that large tropical eruptions are usually followed by a warm sea-surface temperature (SST) anomaly in the Pacific (e.g., Adams et al. , 2003; D'Arrigo et al. , 2005; Li et al., 2013; S. McGregor et al. , 2010; Wilson et al., 2010) even if there are still uncertainties on the significance of theses results (Dee et al., 2020). In addition, El Niño events followed in the first or the second winter after the five largest eruptions of the last 150 years (Krakatau in August 1883, Santa Maria in October 1902, Agung in February 1963, El Chichón in March 1982 and Pinatubo in June 1991). However, the Santa Maria, El Chichón and Pinatubo eruptions occurred after an El Niño event was already developing making it difficult to determine a causal link between ENSO and these eruptions (e.g., Self et al., 1997; Nicholls, 1988).

Moreover, modelling studies initially found divergent responses for the ENSO changes after large tropical eruptions (e.g. Ding et al., 2014; McGregor & Timmermann, 2011; Stenchikov et al., 2006; Zanchettin et al., 2012). However, the majority of the studies have pointed to an El Niño-like response following volcanic eruption (for a review of these studies see Shayne McGregor et al., 2020). In particular, the use of relative sea surface temperature (RSST) or sea surface height (SSH) instead of SST have helped to disentangle the ENSO response from volcanically induced cooling in the Pacific and to highlight the dynamical ENSO response (Khodri et al., 2017; Maher et al., 2015). In this study, the terms El Niño-like and La Niña-like conditions are used to describe an anomalous warming or cooling of the equatorial Pacific relative to a climatology and the term relative is used to describe deviations from the tropical average (20°N–20°S).

Although a consensus is emerging, different aerosol spatial distributions may give rise to different ENSO responses. Stevenson et al., (2016) investigated the impact of northern hemisphere (NH), southern hemisphere (SH) and tropical volcanic eruptions using the Community Earth System Model Last Millennium Ensemble (CESM-LME). They concluded that while NH and tropical eruptions tend to favour El Niño-like conditions, SH eruptions enhance the probability of La Niña-like events within one year following the eruptions. Conversely, Liu et al., (2018) through a millennium simulation performed with CESM 1.0 and Zuo et al., (2018) using the CESM-LME concluded that SH, NH and tropical eruptions all resulted in El Niño-like conditions in the second year after the eruption.

The mechanisms that trigger a change in the ENSO state following volcanic eruptions are still debated. The center of the argument is explaining how a volcanic eruption weakens or amplifies the trade winds (i.e. constant surface eastward winds within the tropics). One of the most frequently used hypotheses is the "ocean dynamical thermostat" mechanism (ODT) (Clement et al., 1996), where a preferential cooling in the western Pacific relative to the eastern Pacific takes place. Such a differential cooling weakens the zonal SST gradient along the equatorial Pacific which causes a relaxation of the trade winds, leading to a temporary weakening of the ocean upwelling in the eastern Pacific. This process is then amplified by the Bjerknes feedback, yielding an El Niño (Bjerknes, 1969). A related mechanism for the preferential El Niño anomalies following volcanic eruptions is based on the recharge-discharge theory of ENSO, including changes in the wind stress curl during the eruption year as one of the triggering factors (McGregor & Timmermann, 2011; Stevenson et al. 2017). However, through a set of sensitivity experiments, Pausata et al., 2020 have questioned the existence of the ODT mechanism in coupled climate models following volcanic eruptions, pointing to the Intertropical convergence zone (ITCZ) displacement and extratropic-to-tropic

teleconnections as key mechanisms in affecting post-eruption ENSO. The ITCZ-shift mechanism was originally proposed for the ENSO response to high-latitude eruptions (Pausata et al., 2015; 2016) and then suggested to also be at work for tropical asymmetric eruptions (Colose et al., 2016; Stevenson et al., 2016). Trade winds converge towards the ITCZ where their intensity weakens, as the air circulates in an upward direction, creating the so-called doldrums (areas of windless waters).

Consequently, a shift of the ITCZ affects the position of the doldrums and the intensity of the trade winds over the equator. In general, the ITCZ shifts away from the hemisphere that is cooled (Kang et al., 2008; Schneider et al., 2014). Consequently, for an eruption with aerosol concentrated in the NH, the ITCZ location moves equatorward, weakening the trade winds and leading to an El Niño-like anomaly via the Bjerknes feedback (Bjerknes, 1969). In contrast, the ITCZ moves northward following a larger SH cooling compared to NH, strengthening the trade winds and triggering La Niña-like anomalies as seen

in Colose et al. (2016) and Stevenson et al. (2016). (Khodri et al., 2017) suggested that the cooling of tropical Africa, following volcanic eruptions, may increase the likelihood of an El Niño events through the weakening of the West African Monsoon and changes in the Walker circulation. More specifically, the authors suggest that the reduction of the tropical precipitation and tropospheric cooling favours anomalous atmospheric Rossby and Kelvin waves in autumn (SON), with a weakening of the trade winds over the western Pacific, leading to El Niño-like conditions in the year after the eruption. A similar mechanism

has been suggested based on the cooling of the Maritime continent or south-eastern Asia instead of tropical Africa (Eddebbar et al., 2019; Ohba et al., 2013; Predybaylo et al.,2017). However, there is yet no consensus as to which of these proposed mechanisms is the main driver of the ENSO response after large volcanic eruptions.

Modeling studies have investigated the impact of volcanic eruptions on ENSO using different approaches. Many studies have used a superposed epoch (SEA) or composite analysis, in which they used a window of a few years before the eruption to

create a reference to compare with the post-eruption period (e.g. Liu, et al., 2018; Zuo et al., 2018). The significance of the response to volcanic eruptions is then assessed using a Monte Carlo method. However, this statistical methodology has some shortcomings as it is not able to fully remove the signal of internal variability: ENSO anomalies can still be seen in the reference period (see for example figure 4 in Liu , et al., (2018)). Some other studies use a small number of ensemble members with volcanic forcing and ensemble members without volcanic forcing starting from different initial conditions (e.g. McGregor &

Timmermann, 2011; Predybaylo et al., 2017; Sun et al., 2019). However, when starting the two ensemble sets from different initial conditions, a large number of ensemble members (equivalent to at least 150 years reference period/climatology) is needed to isolate the internal variability of ENSO (Milinski et al., 2019; Wittenberg, 2009). A recent study by Predybaylo et al., 2020 additionally demonstrates the need for using large ensembles to investigate the ENSO response to volcanoes, utilising 54 100-member ensembles of idealised experiments, they found that the role of the initial ENSO state as well as decomposing

the response into a deterministic and stochastic component are important specially to highlight the role of the ODT mechanism. Here, for the first time, we use a 200-member ensemble taken from the Max Planck Institute Grand Ensemble (MPI-GE) for the historical simulations (1850-2005) (Maher et al., 2019) to investigate the ENSO response to hemispherically symmetric

and asymmetric volcanic eruptions. The large ensemble and the different aerosol distributions allow us to shed light on the mechanisms at play in altering the ENSO state after volcanic eruptions.

## 2 Methodology and Experimental Design

In this study we consider the three largest eruptions, in terms of quantity of aerosols injected in the stratosphere, of the last 100 years; the Agung in Indonesia (8 °S 115 °E) in February 1963; El Chichón in Mexico (17 °N 93 °W) in March 1982; and the Pinatubo in the Philippines (15 °N 120 °E) in June 1991. Two eruptions have an asymmetrical aerosol distribution, so either the aerosols are confined to the NH (El Chichón) or to the SH (Agung). The Pinatubo eruption has an approximately symmetrical distribution with the sulfate aerosols spread across both hemispheres. We also considered the eruption of Krakatau in Indonesia (6 °S 105 °E) in 1883, which is also modelled with an approximately symmetrical aerosol plume to further corroborate our results. For clarity, we only show the results for Pinatubo, which are qualitatively similar to Krakatau. The results of Krakatau are displayed in the supplementary material (Figs. A7, A8 and A9). The stratospheric aerosols used in this study are prescribed in the historical simulations of the model (MPI ESM) from the data set of Stenchikov et al. (1998) (Giorgetta et al., 2013). The data set consists in monthly averages of the radiative properties of the aerosols such as the single scattering albedo, the aerosols extinction and the asymmetry factor and they are based on satellite observation of Pinatubo's eruption (Schmidt et al., 2013).

We use 200 ensemble members of the historical simulations (1850-2005) performed using the Max Plank Institute for Meteorology Earth System Model 1.1 (MPI-ESM 1.1) (Giorgetta et al., 2013) as part of the Max Planck Institute Grand Ensemble (Maher et al., 2019). Such a large ensemble also allows us to analyse the ENSO response of individual eruptions instead of a composite of multiple eruptions as done in some previous studies that used a small number of ensembles (e.g. Maher et al., 2015; Stevenson et al., 2016; Zuo et al., 2018). All the ensemble members are initialized from different years of a long preindustrial control run (2000 years) after it has reached a quasi-stationarity state. The model horizontal resolution is roughly 1.8° for the atmosphere and 1.5° for the ocean with 16 vertical levels for both the atmosphere and the ocean. The MPI-ESM has been extensively used to investigate the impacts of volcanic eruptions on climate (e.g. Bittner et al., 2016; Illing et al., 2018; Timmreck et al., 2016). In particular our model is able to reproduce the global cooling of around 0.2 °C following the three eruptions investigated in this study as well as the high-latitude winter warming (locally up to 2°C) after El Chichón and Pinatubo in the first winter after the eruption (see Fig. A18) in agreement with the estimates provided in the literature (e.g. Robock & Mao, 1995; Timmreck, 2012). Furthermore, while it has been argued that models need a sufficiently high top and good vertical resolution are necessary to properly represent many of the surface climate effects of volcanic eruptions (e.g., Driscoll et al., 2012), (Suarez-Gutierrez et al., 2021) analysed 10 different global climate model and found that MPI-GE was the best model in representing both global and regional internal variability and forced response. Therefore, the limited number of vertical levels does not translate into a poor representation of the forced response associated to volcanic eruptions.

The anomalies calculated are the difference between the reference (3 years before the eruption and for the 200 ensembles members that we refer to as climatology) and periods after the eruption. A Student t-test is used to estimate the significance of the mean changes before and after the eruptions at the 95% confidence level.

Large tropical eruptions induce a global cooling so that El-Niño response may be partly masked, and the La-Niña response amplified (Maher et al., 2015). Furthermore, some climate models overestimate the volcanically induced cooling (e.g. Anchukaitis et al., 2012; Stoffel et al., 2015). To better highlight the dynamical changes, we remove the tropical SST mean from the original SST, this is known as the relative sea surface temperature (RSST) (Vecchi & Soden, 2007). In this study, we use the RSST to isolate the intrinsic ENSO signal, but the analysis using the SST are available in the appendix (Khodri et al., 2017).

## 3 Results

### 3.1 ENSO response and its links to the ITCZ-shift mechanism

The volcanic eruptions analyzed in the present study show three distinct aerosol plumes. While the aerosol distribution from the Pinatubo eruption is symmetrical around the equator, Agung and El Chichón eruptions both created, to a large extent, a confined distribution in the SH and the NH respectively (Fig. 1). For the Pinatubo and El Chichón the aerosol optical depth peaks in the winter that follows the eruptions. For the Agung this maximum is reached in the winter of the second year after the eruption. The reasons why similar eruptions can lead to different aerosol distributions are still being investigated, but some causes are the location of the volcanoes, the season and the strength of the eruption (Stoffel et al., 2015; Toohey et al., 2011), the Quasi-Biennial Oscillation and local meteorology (e.g. Choi et al., 1998), which determine wind direction and the aerosols distribution across the equator. Regarding the Pinatubo aerosol distribution, we note that the eruption was followed few months after by the eruption of Cerro Hudson (Chile) in the Southern Hemisphere. As the aerosol distribution for Pinatubo is based on satellite observations, the aerosol properties prescribed to the model also include Cerro Hudson eruption

Our simulations show a relative El Niño-like response to the El Chichón and Pinatubo eruptions and a relative La Niña-like response to the Agung eruption (Figs. 2 and 3). The relative ENSO anomalies develop at the beginning of the year after the eruption (Year 2) and then peak that boreal winter for El Chichón and Pinatubo or in the third boreal winter for Agung. The westerly anomalies in the trade winds are detected starting in the autumn of the eruption (Year 1). For all the eruptions, the relative Niño 3.4 index peaks in the winter of the year after the eruption reaching a maximum of approximately +0.3 °C for El Chichón and the Pinatubo and a minimum of -0.2 °C for Agung (Fig. A3). The results using the SST (Figs. A11, A12 and A13) are qualitatively similar to the RSST (Figs. 2, 3, and A3), but the El Niño-like anomalies are less strong since they are partially masked by the global cooling induced by the stratospheric aerosols, which is in agreement with other studies (Khodri et al., 2017; Maher et al., 2015).

Furthermore, the temperature anomalies following the Agung eruption are not just superficial, but extend down to 200 m depth (Fig. A14). The warming below 100 m in the western Pacific and the cooling along the thermocline in the eastern side of the

basin are typical of an ongoing La Niña : the pattern is indeed remarkably similar to a composite of La Niña events occurring in the reference periods before the three eruptions (Fig. A17). On the other hand, the temperature anomalies for the El Chichón and Pinatubo eruptions show opposite results as expected under El Niño-like conditions (Figs. A15 and A16). Moreover, the wind anomalies following Agung (see Fig. 2 and 3) are opposite compared to El Chichón, hence leading to opposite upwelling conditions along the equator, further corroborating that the ocean temperature anomalies are mostly dynamically driven through the Bjerknes feedback (Bjerknes, 1969). The different forcing caused by the three eruptions induces a different cooling of the surface temperature in the two hemispheres (Fig. 4). Although Pinatubo is the most intense eruption and has the largest global temperature decrease (Fig. A1), El Chichón shows the strongest hemispherical cooling (Fig. 4). The relative hemispherical cooling associated to Agung (SH) is in absolute values comparable to Pinatubo even if Agung's quantity of aerosols injected in the stratosphere was twice as small (Bluth et al., 1992; Self & Rampino, 2012). Furthermore, while the aerosol distribution of the Pinatubo eruption is symmetric, the relative cooling is not and is concentrated in the NH, which is likely due to uneven distribution of landmass between hemispheres (i.e. reduced heat capacity in the NH). The maximum cooling for all three eruptions occurs in the second year and so does the temperature difference between the two hemispheres (Figs. 4 and A1).

Precipitation anomalies (Figs. 5 and 6) show the displacement of the Hadley cell associated with the differential cooling between the hemispheres, showing a northward shift of the ITCZ for the eruption of the Agung, and a southward shift for both El Chichón and the Pinatubo, for the three years following the eruption. Additionally, Pinatubo's rainfall anomaly is the largest even though the difference in the cooling of each hemisphere is stronger for El Chichón. This indicates that the ITCZ response is sensitive to the magnitude and spatial distribution of the forcing. The on-going global warming could also amplify the rainfall anomaly following the volcanic eruptions through modulation in the ocean stratification and near-surface winds amplifying the response as suggested in a recent study (Fasullo et al., 2017) (Fig. A9). Moreover, the precipitation asymmetry index (PAi) (Colose et al., 2016) further highlights the ITCZ shift and weakening (Fig. 7). The expression for the PAi is given by:

$$PAi = \frac{P_{20°N-10°N} - P_{10°N-Eq}}{P_{20°N-Eq}}$$

It represents the variation of the zonal precipitation around the average climatological mean of the ITCZ in our model, which is roughly 10 °N. In figure 7, the positive variation of the PAi after Agung eruption corresponds to a northward shift of the ITCZ and the negative variations of the PAi after Pinatubo and El Chichón eruptions corresponds to a southward shift of the ITCZ.

We find that the ITCZ displacement and associated rainfall anomalies peak the second year after the eruption, when the differential cooling of the hemisphere is larger (Figs. 4, 5, 6 and 7). The ITCZ displacement is associated with a strengthening of the trade winds for the Agung eruption and a weakening for the El Chichón and Pinatubo eruptions as expected by the direction of the ITCZ movement in each case. These wind anomalies then affect the ENSO state: a change in the strength of the trade winds along the equatorial Pacific alters the ocean upwelling in the eastern side. This leads to a change in the east-

west temperature contrast across the tropical Pacific, which is amplified by the Bjerknes feedback thus altering the ENSO state. All our results (the evolution of the relative Niño 3.4 index, the precipitation anomalies or the relative temperature anomalies) show an almost perfect symmetry between the tropical/NH distribution and the SH distribution (Figs. 2, 3, 4, 5, 6 and A2), which strongly suggests that the volcanically induced ITCZ displacement is the key mechanism to explain the ENSO response to the volcanic forcing in agreement with others studies (Colose et al., 2016; Pausata, et al., 2016; Stevenson et al., 2016; Pausata et al,. 2020).

## 3.2 ENSO response and its link to other mechanisms

The most commonly invoked mechanism that explains the ENSO response after large tropical volcanic eruptions is the ODT (Clement et al., 1996) and the preferential cooling of the warm pool relative to the eastern equatorial Pacific, leading to an El Niño-like response (e.g. Emile-Geay et al., 2008; Mann et al., 2005). However, in our simulations even if there is a volcanic aerosol over the equatorial Pacific and a surface cooling for all the eruptions (Figs. 1, A3, A4 and A5), we see a negative phase of the relative ENSO and an anomalous easterly wind stress developing after the Agung eruption (Figs. 2 and 3). Should the ODT mechanism be dominant in our model, we would expect the opposite response of both the relative SST and wind stress. This suggests that the ODT is likely not the leading mechanism in our model, while not ruling out that it can still play a role under specific initial conditions (e.g. Predybaylo et al., 2020). Another mechanism often used to explain the post-eruption El Niño-like response is related to the cooling of the Maritime Continent first proposed by Ohba et al., 2013 and also suggested in recent studies (e.g. Eddebbar et al., 2019) : the smaller heat capacity of the land in comparison to the ocean cause a stronger land cooling that reduces the temperature gradient between the Maritime Continent and the western Pacific Ocean. Such temperature changes lead to a weakening of the trade winds and an eastward shift of the rainfall (El Niño pattern). Our results show a cooling of the Maritime Continent and a reduction of the convective activity in the three eruptions (Figs. A3, A4 and A5). Nevertheless, our model simulates the development of a relative La Niña-like conditions after the eruption of the Agung (Figs. 1 and 2), which is at odds with the cooling of the Maritime Continent mechanism, where relative El Niño-like conditions would be expected for all three eruptions.

Khodri et al., (2017) suggested that the cooling of tropical Africa, following volcanic eruptions, may increase the likelihood of an El Niño events through the weakening of the West African Monsoon and changes in the Walker circulation. More specifically, the reduction of the tropical precipitation and tropospheric cooling favours anomalous atmospheric Rossby and Kelvin waves in autumn (SON), with a weakening of the trade winds over the western Pacific, leading to El Niño-like conditions in the year after the eruption. Our results show a mixed response over Africa with Agung and Pinatubo both displaying a cooling (Figs. A3 and A5) but leading to different relative ENSO responses (Figs. 1 and 2). Moreover, after El Chichón eruption a warming of the tropical northern Africa takes place in the first year and an El Niño-like anomaly develops (Figs. 2, 3 and A4). Hence, in our model, the volcanically induced cooling of tropical Africa and the atmospheric perturbations associated with the suppression of the African monsoon seem not to play a critical role in altering the ENSO state following volcanic eruptions.

Recently, Pausata et al., 2020 proposed an additional mechanism related to the extratropical-to-tropical teleconnections that tends to favour an El Niño-like response for both NH and SH eruptions, hence playing in synergy (NH eruptions) or against (SH eruptions) the ITCZ-shift mechanism. However, our qualitative analysis of the sea level pressure (SLP) anomalies does not match the changes expected by this mechanism (cf. Fig. A10 to Fig. 4 (a-d) in Pausata et al., (2020)). In this recent study, the volcanic aerosol alters the meridional temperature gradient of the atmosphere that eventually causes a poleward shift of the Pacific jet stream and a strong cyclonic surface pressure anomaly over the midlatitude to subtropical Pacific basin in both NH and SH eruptions in the first summer following the eruptions. In our simulation, the response is opposite for the Agung and El Chichón or Pinatubo eruptions, suggesting more that the simulated extratropical anomalies are induced by the relative ENSO changes due to the eruption rather than affecting ENSO (Fig. A10). The reason of the disagreement could lie in the fact that the El Niño/La Niña-like response following the volcanic eruptions peak in the first winter in Pausata et al., (2020) modeling study, while in our model in the second winter (Figs. 2 and 3). The extratropical-to-tropical teleconnection could make the El Niño development following NH/symmetric eruptions occur faster than in our case where such a teleconnection appears not to be present. Ad hoc sensitivity experiments are necessary to rule out the above-mentioned mechanisms in our model.

## 4 Discussion and conclusions

Our study used the largest ensemble simulation (200 ensembles) currently available of the historical period performed with the MPI-ESM model to better understand the impact of the volcanic eruptions on ENSO. Our results strongly point to the volcanically induced ITCZ displacement as the primary driver of the ENSO response following volcanic eruptions in this model. In our simulations, the ENSO response after the eruptions critically depends on the distribution of the aerosol plume. When the volcanic aerosol distribution is confined to the NH or its distribution is symmetrical across the hemispheres the ENSO state tends towards a positive phase (relative El-Niño like conditions; Fig. 2 (d-i)), while when the aerosols are confined to the SH the ENSO state is pushed towards a negative phase (relative La Niña-like conditions; Figs. 2 (a-c)). The displacement of the ITCZ following the eruptions, caused by the asymmetric cooling of the hemisphere that pushes the ITCZ towards the hemisphere that is less cooled (Kang et al., 2008; Schneider et al., 2014). Both the eruptions with aerosol confined to the NH and symmetrically distributed across the hemispheres preferentially cool the NH, consequently shifting the ITCZ southwards, weakening the trade winds over the equatorial Pacific and triggering El Niño like response through the Bjerknes feedback (Bjerknes, 1969). The eruption with the aerosol plume confined to the SH instead cools exclusively the SH, pushing the ITCZ northward and strengthening the trade winds, leading to La Niña-like response (Figs. 5, 6 and 7).

The ITCZ mechanism we see at play in our model is supported by other recent studies performed with different climate models (Pausata,et al., 2015, 2020; Stevenson et al., 2016; Colose et al., 2016). Pausata et al. (2020) through a set of sensitivity experiments in which the volcanic aerosol forcing is confined to either the NH or the SH show the key role of the ITCZ displacement in driving the ENSO response. They also highlighted the presence of another mechanism related to the

extratropical-to-tropical teleconnections that no matter the type of eruption (NH or SH) tends to favour an El-Niño like response. Hence, it plays in synergy (NH eruptions) or against (SH eruptions) the ITCZ-shift mechanism. However, the simulated SLP changes in the extratropics in our model seem to be in response to the volcanically induced ENSO changes rather than affecting the ENSO response (cf. Fig. A10 to Fig. 4 (a-d) in Pausata et al., (2020)).

Our work also pointed out that the ODT (Clement et al., 1996), the cooling of the Maritime Continent (Ohba et al., 2013) and of the tropical Africa (Khodri et al., 2017) are likely not the dominant drivers of the ENSO response in the MPI-GE model. Discrepancies between these previous studies and our results may occur for three reasons. First, different models may present different mechanisms. The fact that the above-mentioned mechanisms are not dominant is also in qualitative agreement with the modeling experiments in Pausata et al. (2020), who use the Norwegian Earth System Model (NorESM). Another study using all the ensemble members (14) of a large ensemble (CESM-LME) is also in overall agreement with our results (Stevenson et al. 2016), finding a La Nina-like response to SH eruption and an El Nino-like response to NH and tropical eruption. However, other studies using either a subset (5 members) of the CESM total ensemble (Zuo et al. 2018) or an early version of CESM with only one ensemble (Liu et al., 2018) display El-Nino like anomalies for all type of eruptions, which brings us to the following point. Second, most previous studies are based on a small number of ensemble members (e. g. 5 members for 3 eruptions for the SH plume in Zuo et al. (2018)) or they heavily rely on statistical tools using a small sample of events (e. g. SEA in Liu et al. (2018)). Consequently, those results may be biased by the use of a restrained number of ensemble members. Here, our study points out the importance of a large number of ensemble members when investigating the ENSO response to volcanic eruptions. Third, the role of the initial ENSO state is important for determining the ENSO response (Pausata et al., 2016; Predybaylo et al., 2017) and consequently it influence the role of the ODT and the other above-mentioned mechanisms. Our study only considers the total climate response to volcanic forcing; however, in a recent study, Predybaylo et al, (2020) separated the ENSO response to volcanic eruptions into a deterministic and stochastic response. They have shown that the deterministic response is dominant for spring and summer eruptions, while stochastic response plays a major role for eruptions occurring in winter. However, our experimental design does not allow the separation of the total climate response into the deterministic and stochastic component with further future experiments needed in which the volcano and the reference no-volcano ensemble members start from the same initial conditions.

Finally, our results are consistent with the predominance of post-eruption El Niño events (Adams et al., 2003; McGregor et al., 2020) and it can provide an explanation on why the majority of both observations and reconstructions are displaying El Niño events instead of La Niña events. However, the ENSO responses discussed in this study are only tendential (El Niño-like or La Niña-like response), i.e., intrinsic variability evolving toward a La Niña at the time of the eruption would not necessarily lead to a post-eruption El Niño event even for a NH or symmetrical eruption, but rather to a dampening of the ongoing La Niña. Furthermore, our model suggests the peak in ENSO anomalies to be in the second or third winter after the eruption similar to most modeling studies (Khodri et al., 2017; Lim et al., 2016; McGregor & Timmermann, 2011; Ohba et al., 2013; Stevenson et al., 2017). This is at odds with the reconstructions and observations that see a peak in ENSO anomalies in the first winter following the eruption (McGregor et al., 2010, 2020) and possibly extending to the second year (Adams et al.,

2003). The delayed ENSO response in our model simulations relative to reconstructions and observations may be related to the apparent lack of extratropical-to-tropical teleconnections (Pausata et al., 2020) that could favour El Niño-like response already on the first winter following the eruption or other biases within the climate models (e.g., double ITCZ).

In conclusion, our results provide further insights into the mechanism driving the ENSO response to volcanic eruptions, highlighting in particular the role of the ITCZ shift. However, further coordinated efforts with specific sensitivity studies are necessary to delve into the other proposed mechanisms and to unravel the difference between modeling studies and reconstructions with regards to the peak of the ENSO response. Given that ENSO is the major leading mode of tropical climate variability, which has worldwide impacts, these types of studies are also necessary to help improve seasonal forecasts following large volcanic eruptions.

**Appendices**

**Appendix A**

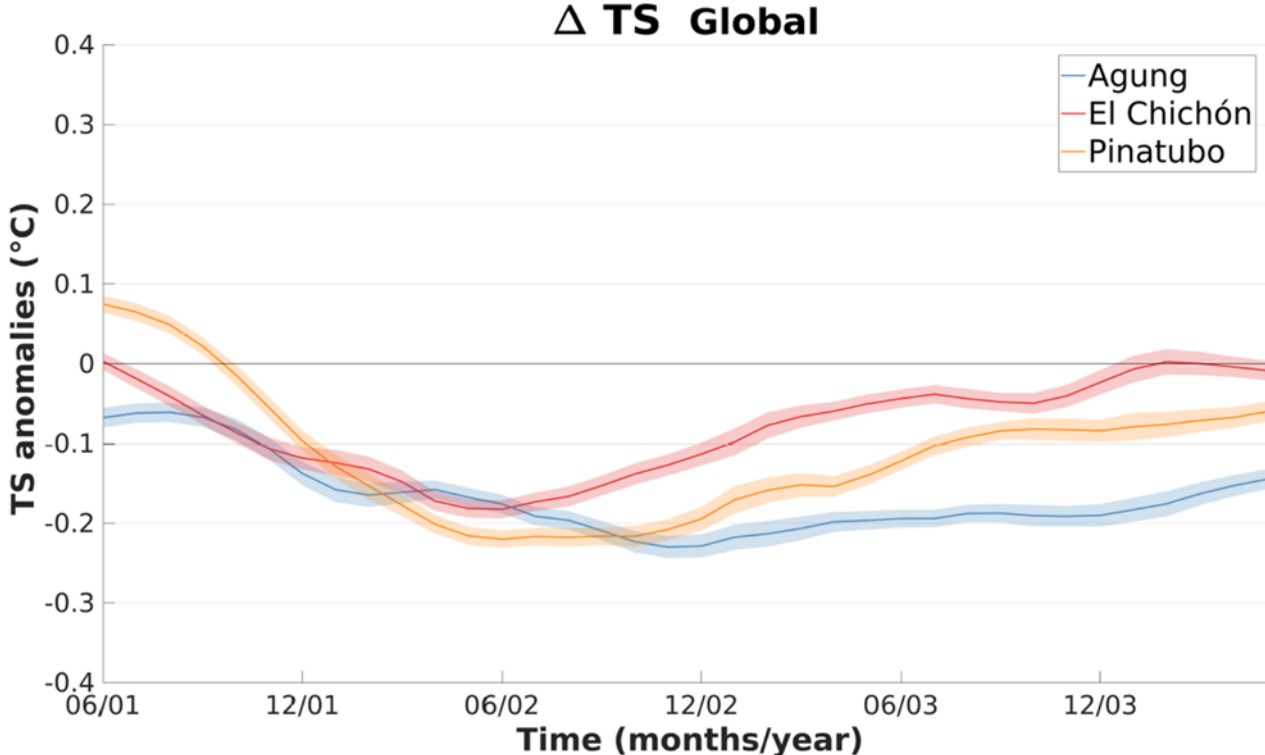

**Figure A1.** Evolution of the ensemble mean of the global the cooling in the three eruptions case for three years, starting at the first summer after the eruption. The 3-year climatology is subtracted to calculate the anomalies. Shading represents twice the standard error of the ensemble mean (i.e. ~95% confidence interval).

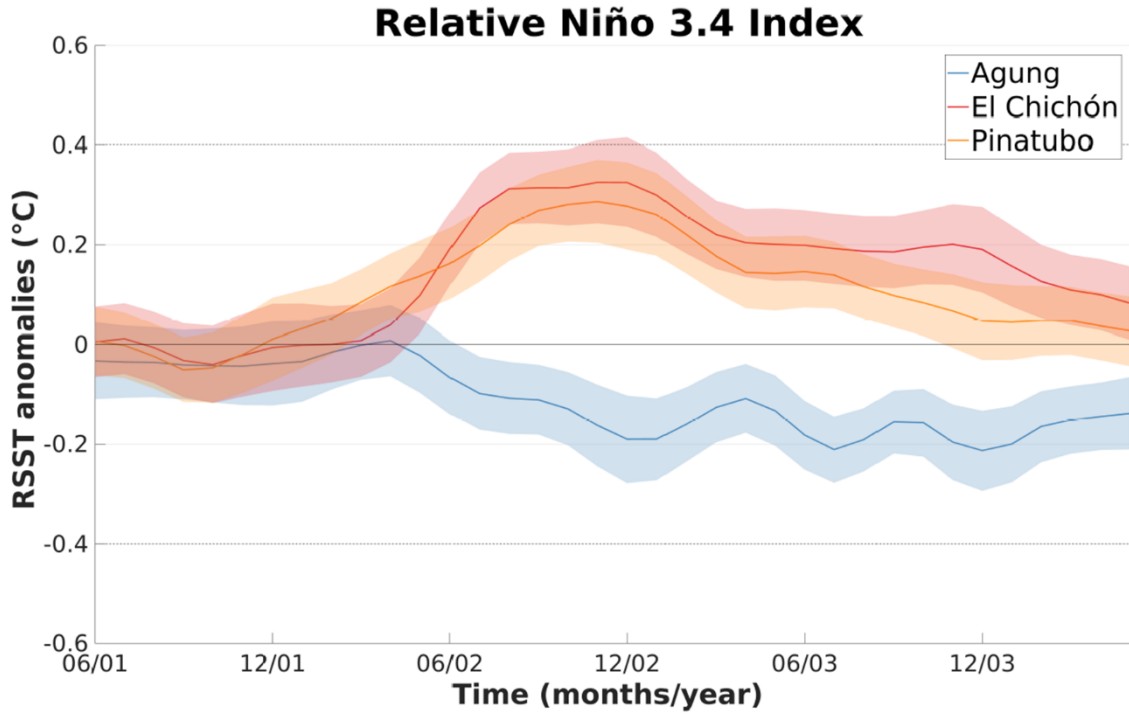


**Figure A2.** Ensemble mean changes in the relative Niño 3.4 index after each eruption. The 3-year climatology is subtracted to calculate the anomalies. Shading represent twice the standard error of the mean using an approximate 95% confidence interval.




# Agung

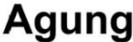

**a) Δ TS** Year 1 (JJA)

**e) Δ PRECIP** Year 1 (JJA)

**b) Δ TS** Year 1 (SON)

**f) Δ PRECIP** Year 1 (SON)

**c) Δ TS** Year 1 (DJF)

**g) Δ PRECIP** Year 1 (DJF)

**d) Δ TS** Year 1 (MAM)

**h) Δ PRECIP** Year 1 (MAM)


**Figure A3.** Ensemble mean of changes in surface temperature (a,b,c,d) and precipitation (e,f,g,h) between the climatology and the volcano case for each seasons of the year after Agung eruption. Only significant anomalies are showed with an approximate 95% confidence level using a Student *t*-test. Contour shows temperature and precipitations anomalies following the color bar scale (solid line for positive anomalies and dashed line for negative anomalies).




# El Chichón

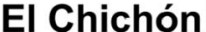

Figure A4. Ensemble mean of changes in surface temperature (a,b,c,d) and precipitation (e,f,g,h) between the climatology and the volcano case for each seasons of the year after El Chichón eruption. Only significant anomalies are showed with an approximate 95% confidence level using a Student *t*-test. Contour shows temperature and precipitations anomalies following the color bar scale (solid line for positive anomalies and dashed line for negative anomalies).

# Pinatubo

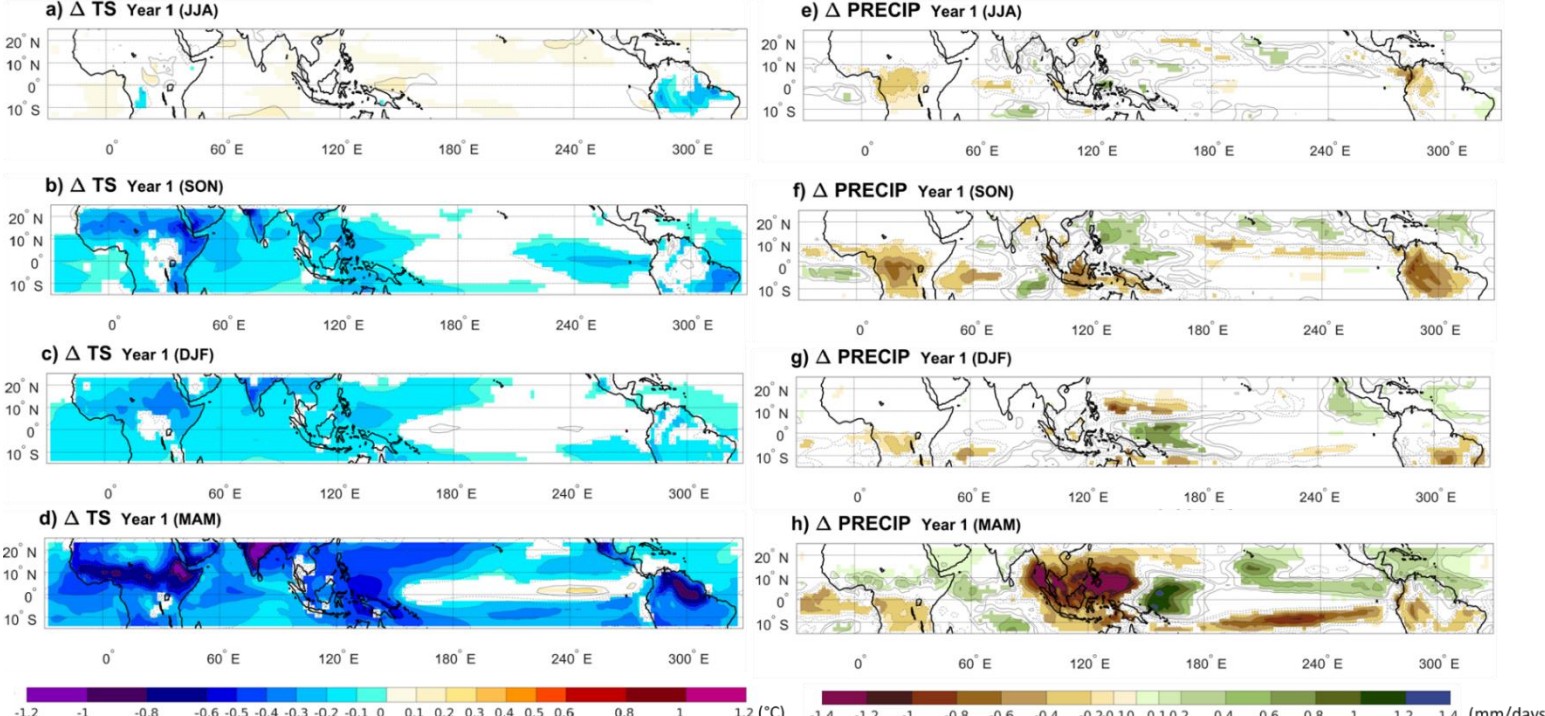

**Figure A5.** Ensemble mean of changes in surface temperature (a,b,c,d) and precipitation (e,f,g,h) between the climatology and the volcano case for each seasons of the year after Pinatubo eruption. Only significant anomalies are showed with an approximate 95% confidence level using a Student *t*-test. Contour shows temperature and precipitations anomalies following the color bar scale (solid line for positive anomalies and dashed line for negative anomalies).




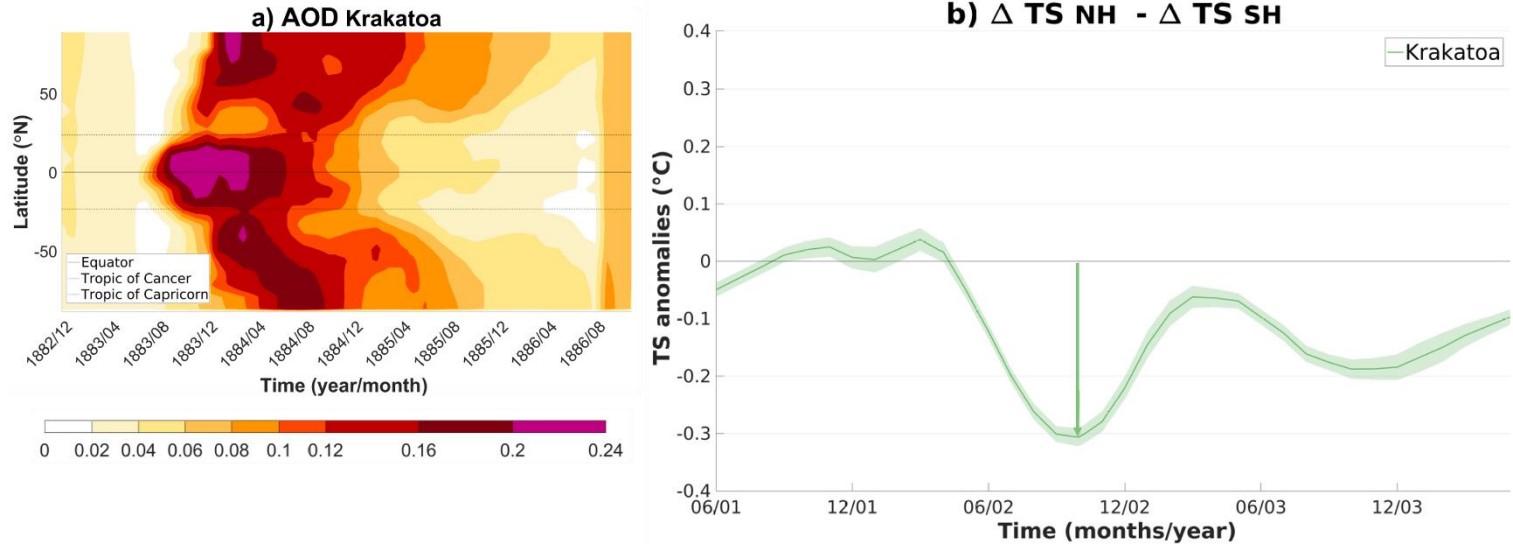

**Figure A6.** Evolution of the aerosol optical depth and ensemble average of the difference between the cooling of the SH (ΔT SH) and the NH (ΔT NH) for the Krakatau eruption. (a) The band of wavelength used is between approximatively 462 nm and 625 nm. (b) 3-year climatology is subtrackedto calculate the anomalies. Shading represents twice the standard error of the ensemble mean (i.e. ~95% confidence

interval).




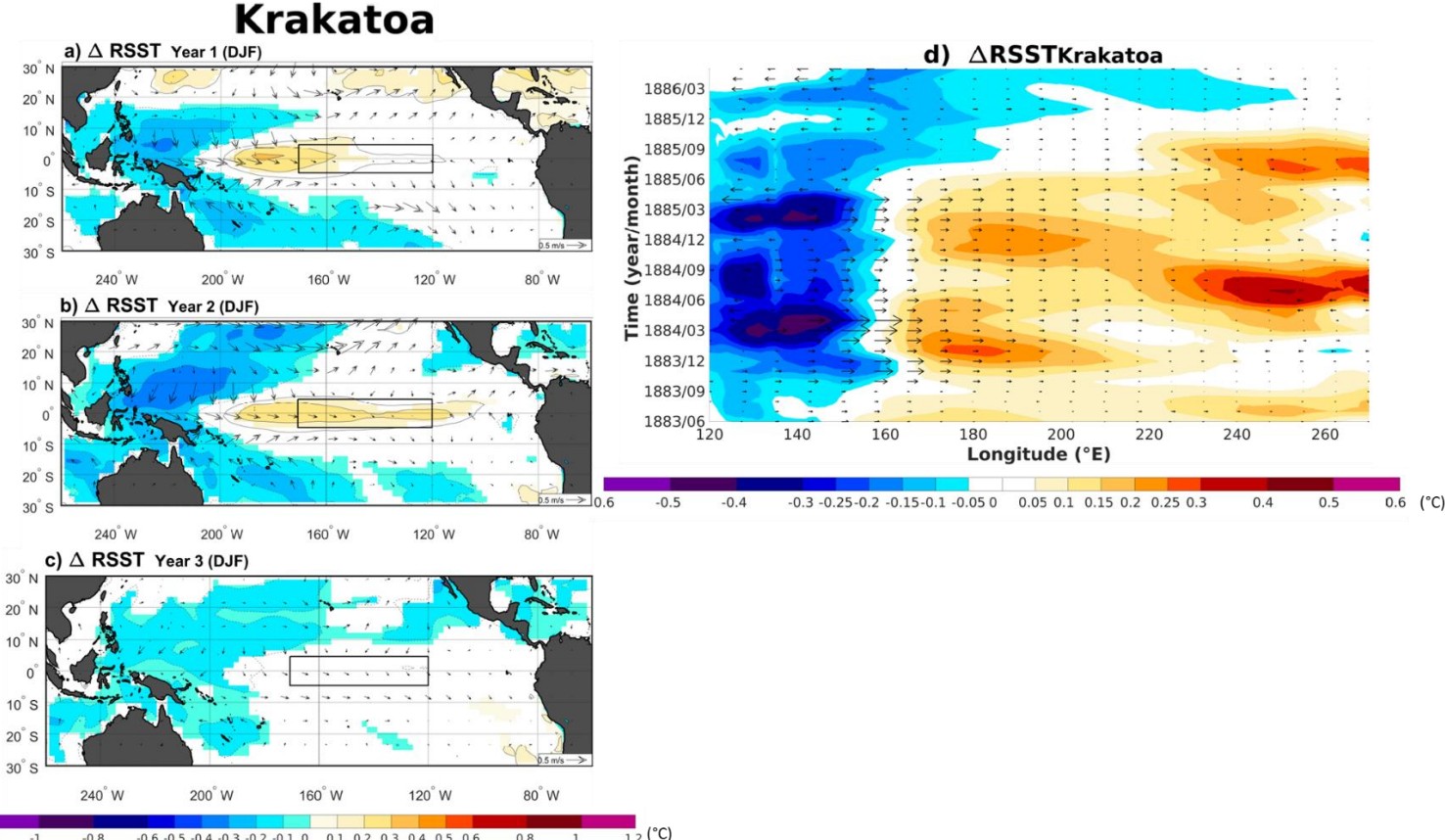

**Figure A7.** Ensemble mean of changes in relative sea surface temperature anomalies and 10m winds (arrows) between the climatology and the volcano case for the three-winter season (DJF) after the Krakatau eruption. Only significant RSST changes are showed with an approximate 95 % confidence level using a Student *t*-test. Contours show the RSST anomalies following the color bar scale (solid lines for positive anomalies and dashed lines for negative anomalies). The boxes indicate the Niño 3.4 area.


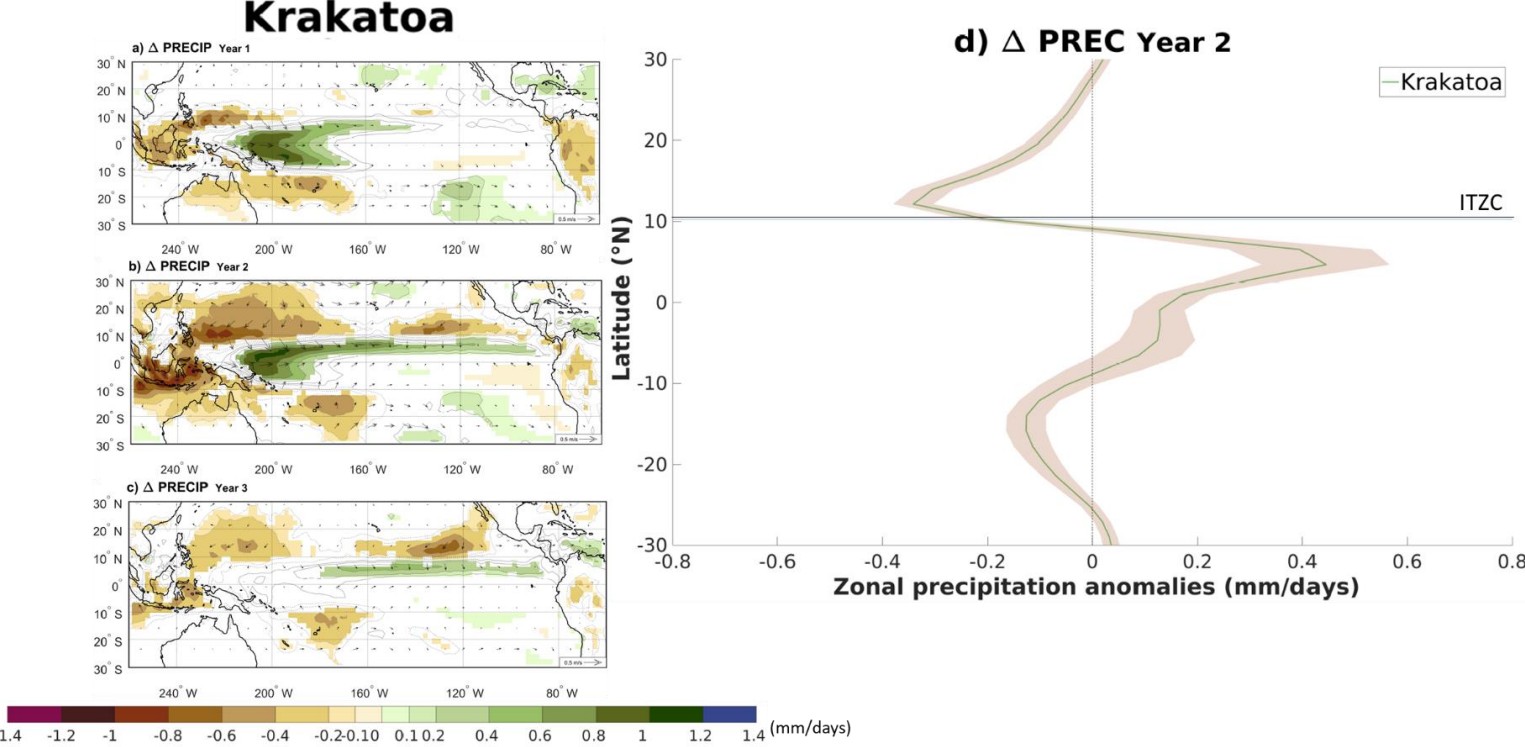

**Figure A8.** (a), (b) ,(c) Ensemble mean of changes in precipitation and 10 m wind (arrows) between the 3-year climatology and the volcano case for the three summer to winter seasons (June to February) following the Krakatau eruption. Only precipitation changes that are significant at the 95% confidence level using a Student *t*-test are shaded. Contours show the precipitation anomaly following the color bar scale (solid lines for positive anomalies and dashed lines for negative anomalies, the 0 line is omitted). (d) Ensemble average of the zonal precipitation anomaly over the Pacific Ocean (160 °E-100 °W) between the 3-year climatology and the summer to winter season (June to February) of the second year after the Krakatau eruption. Shading represents twice the standard error of the ensemble mean (i.e.  ̴95% confidence interval). The blue line highlights the ensemble-averaged 3-year climatology position of the ITCZ (defined as the location of the zonal-average precipitation maximum).

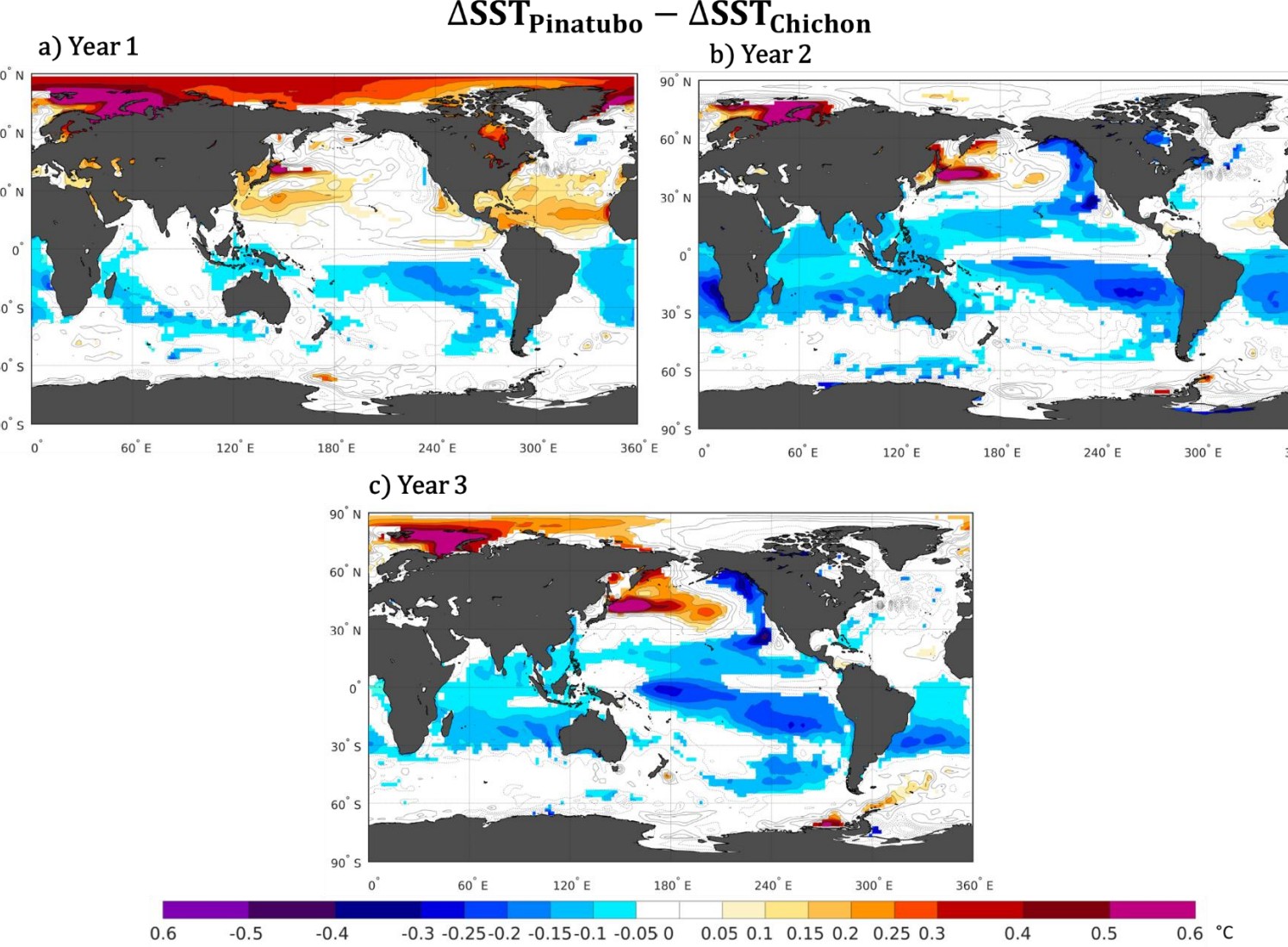

**Figure A9.** Difference in the ensemble mean sea surface temperature anomalies between the Pinatubo and El Chichón eruptions. ($\Delta TS_{Pinatubo}$-$\Delta TS_{Chichón}$). Only significant anomalies are showed with an approximate 95% confidence level using a Student *t*-test. Contour shows temperature anomalies following the color bar scale (solid line for positive anomalies and dashed line for negative anomalies).

# $\Delta\text{SLP}_{\text{Year 1}}$ (JJA)

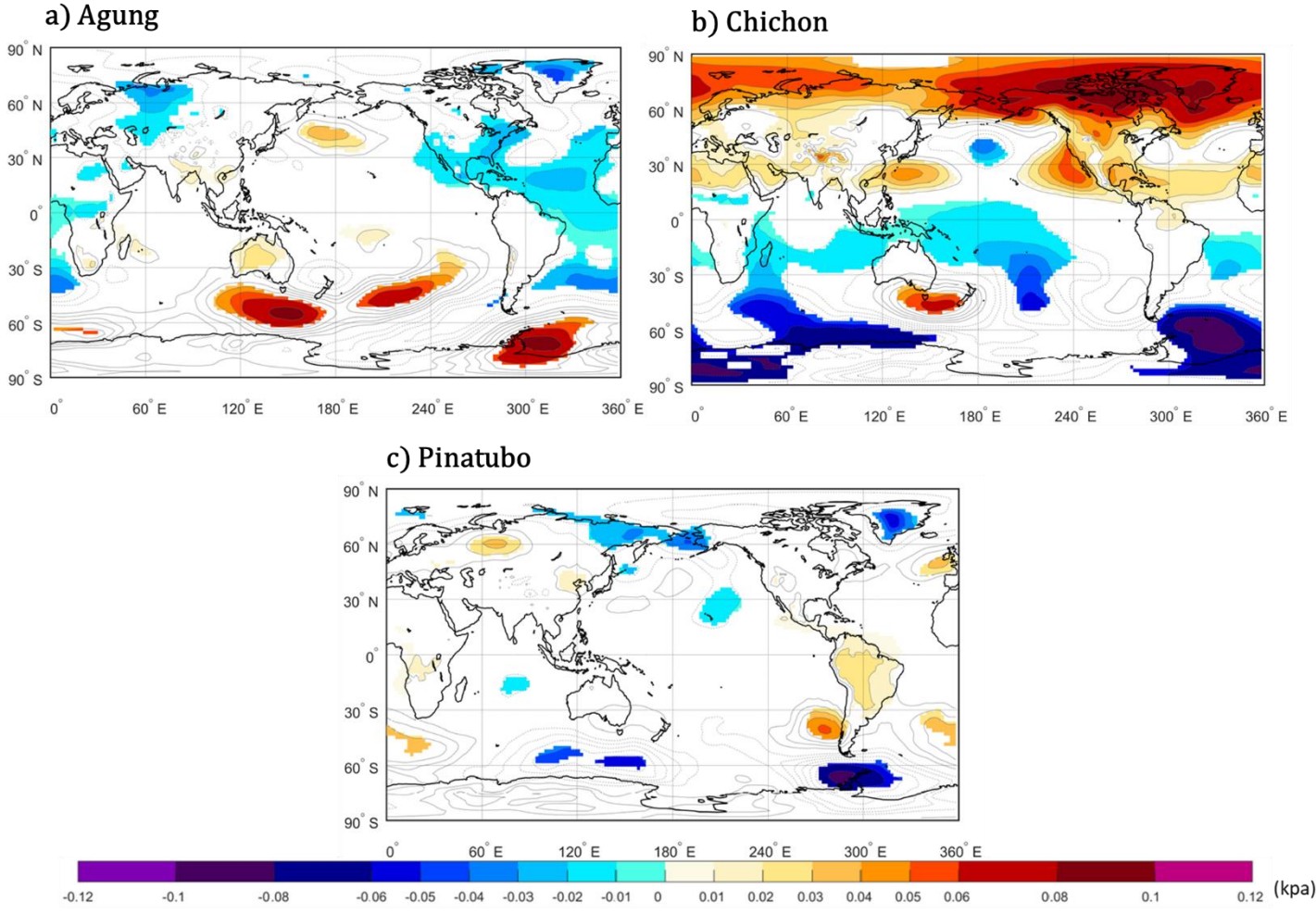

**Figure A10.** Ensemble average of change in the sea level pressure between the climatology and the volcano case for the first summer after Agung (a), El Chichón (b) and Pinatubo (c) eruptions. Only significant anomalies are showed with an approximate 95% confidence level using a Student $t$-test. Contour shows SLP anomalies following the color bar scale (solid line for positive anomalies and dashed line for negative anomalies).



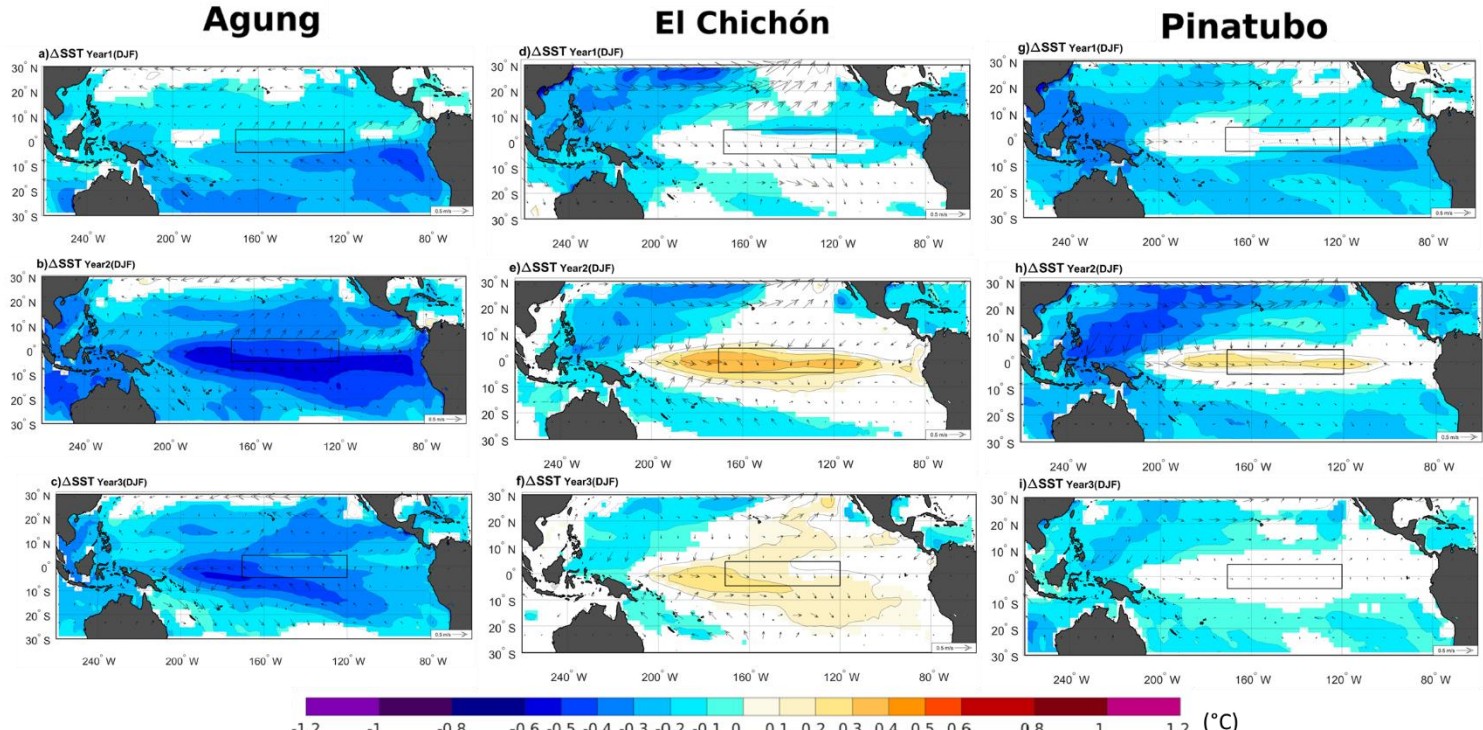

**Figure A11.** Ensemble mean of changes in sea surface temperature (SST) (shadings) and 10 m winds (arrows) between the volcano case and the climatology for each of the following three winter season (DJF) after the Agung (a-c), El Chichón (d-f) and the Pinatubo (g-i) eruptions. Only significant SST changes are shaded with an approximate 95 % confidence level using a Student t-test. Contours show the SST anomalies following the color bar scale (solid lines for positive anomalies and dashed lines lines for negative anomalies, the 0 line is omitted). The boxes indicate the Niño 3.4 area.

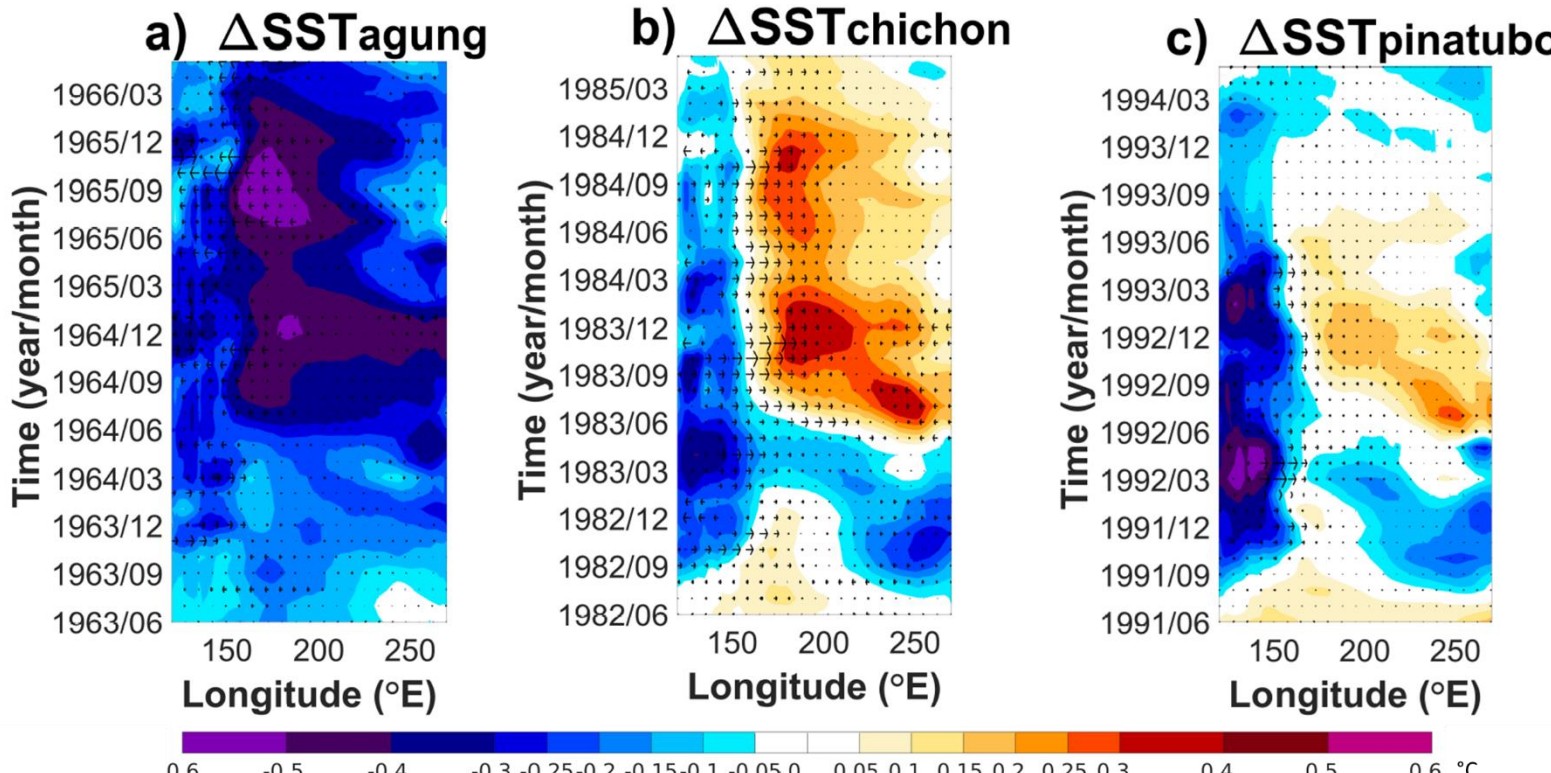

**Figure A12.** Hovmöller plot of the ensemble mean of the SST anomalies in the equatorial Pacific (averaged over -5 °N and 5 °N) and the change in the zonal component of the 10 m winds (m/s) for the three years following each eruption. The anomalies are calculated relative to the three years before each eruption.



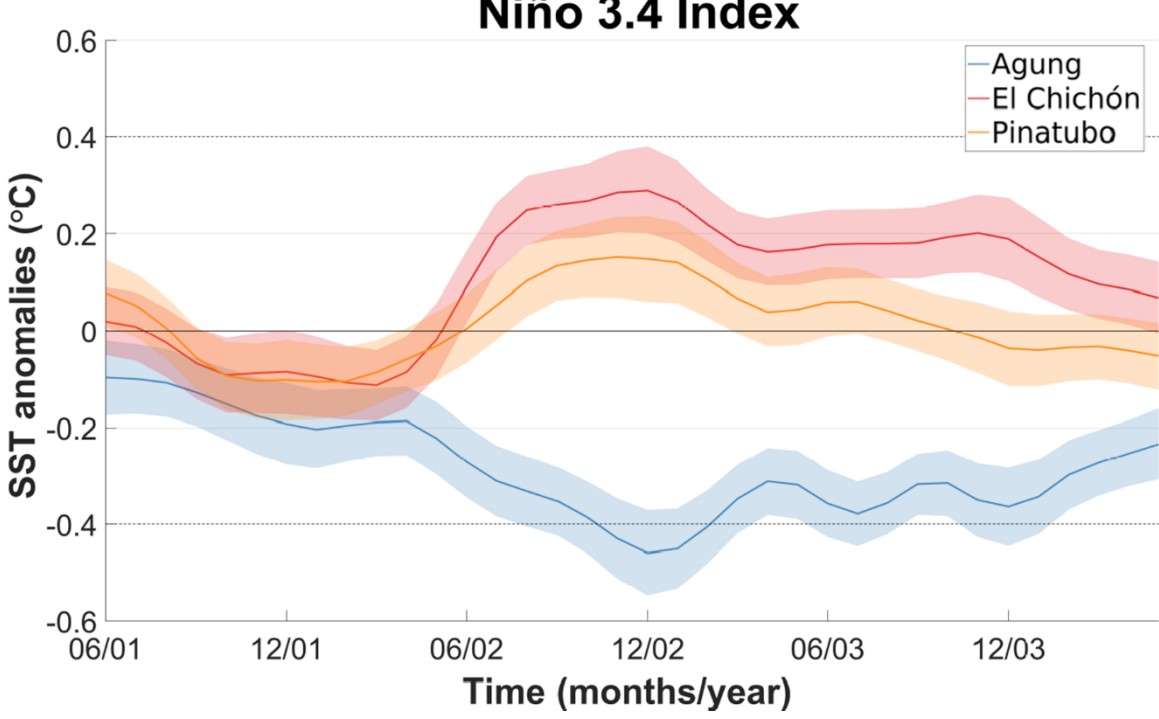

**Figure A13:** Ensemble mean changes in the Niño 3.4 index after each eruption. The 3-year climatology is subtracted to calculate the anomalies. Shading represent twice the standard error of the mean using an approximate 95% confidence interval.




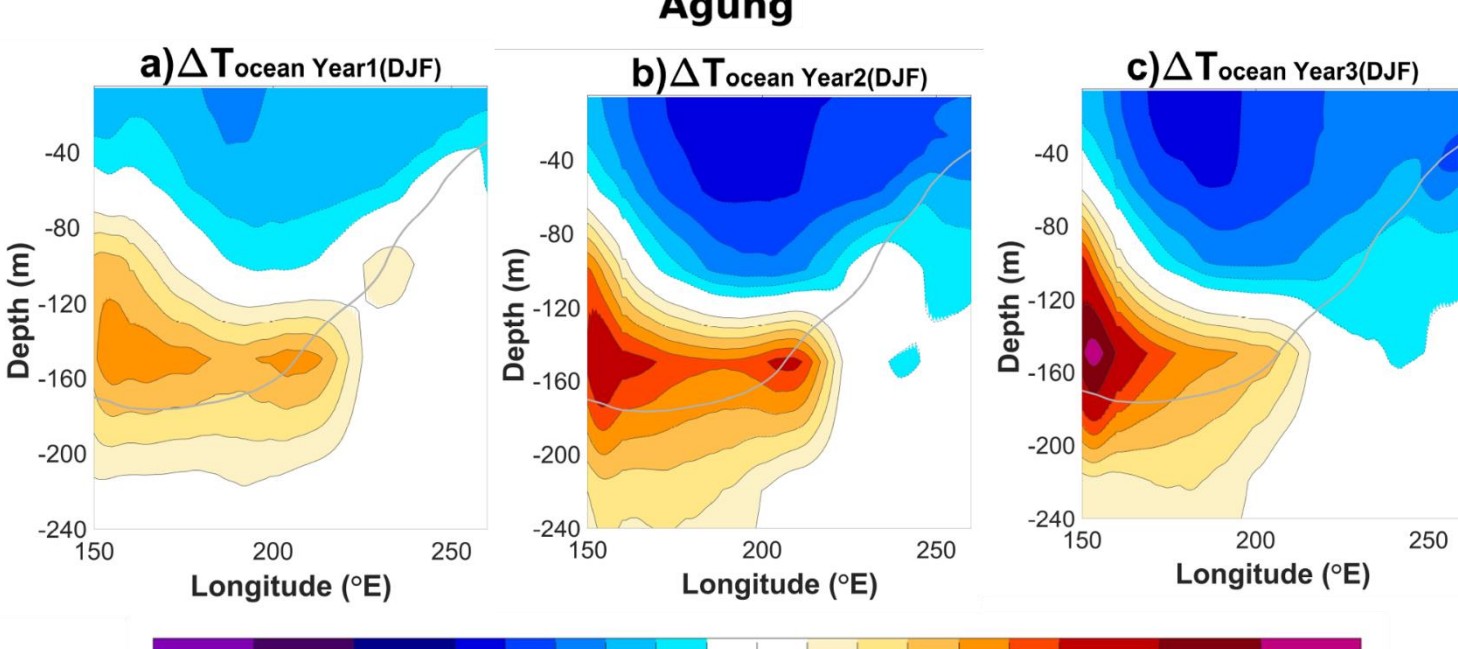

**Figure A14.** Ensemble mean changes shown for a transect in the equatorial Pacific (averaged 5°N – 5°S) of the ocean temperature (shadings)
between the volcano case and the climatology for each of the following three winter season (DJF) after the Agung eruption. Contours show
the SST anomalies following the color bar scale (solid lines for positive anomalies and dashed lines lines for negative anomalies, the 0 line
is omitted). The bold grey line shows the climatological thermocline depth (as defined using the 20°C isotherm). This is shown for 100
ensemble members.





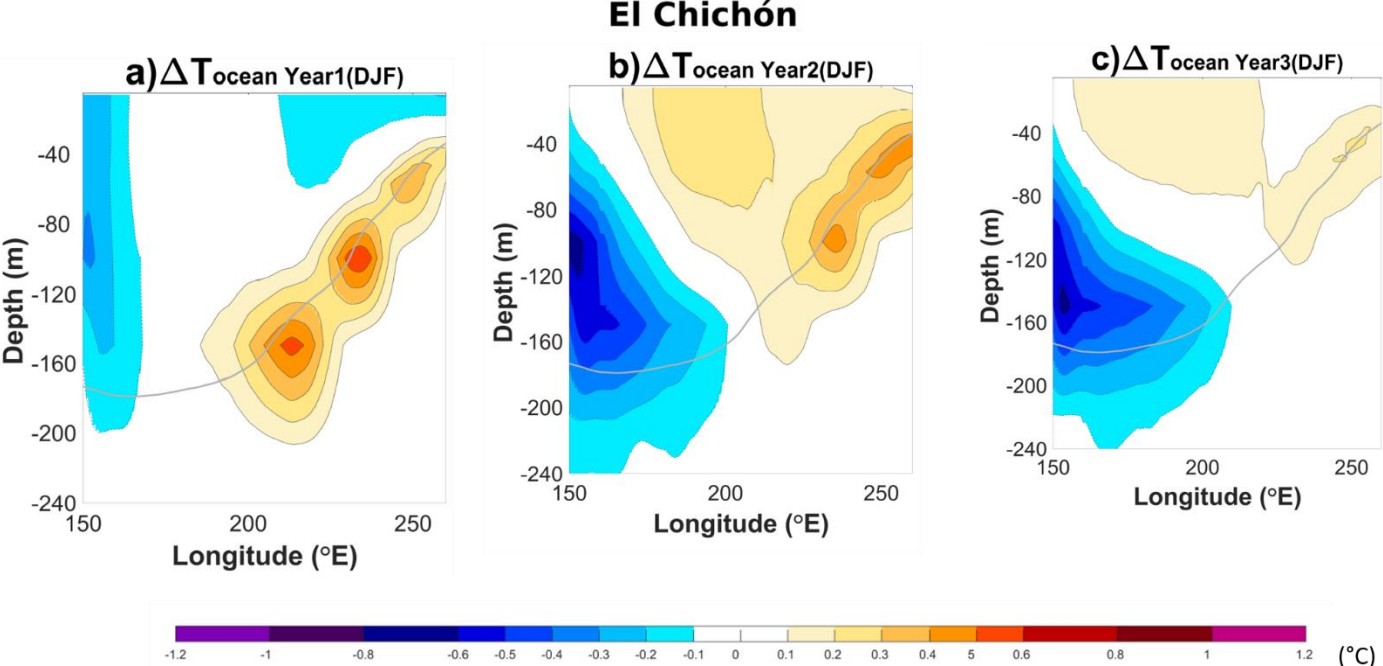

**Figure A15.** Ensemble mean changes shown for a transect in the equatorial Pacific (averaged 5°N – 5°S) of the ocean temperature (shadings) between the volcano case and the climatology for each of the following three winter season (DJF) after the Chichón eruption. Contours show the SST anomalies following the color bar scale (solid lines for positive anomalies and dashed lines lines for negative anomalies, the 0 line is omitted). The bold grey line shows the climatological thermocline depth (as defined using the 20°C isotherm). This is shown for 100 ensemble members.


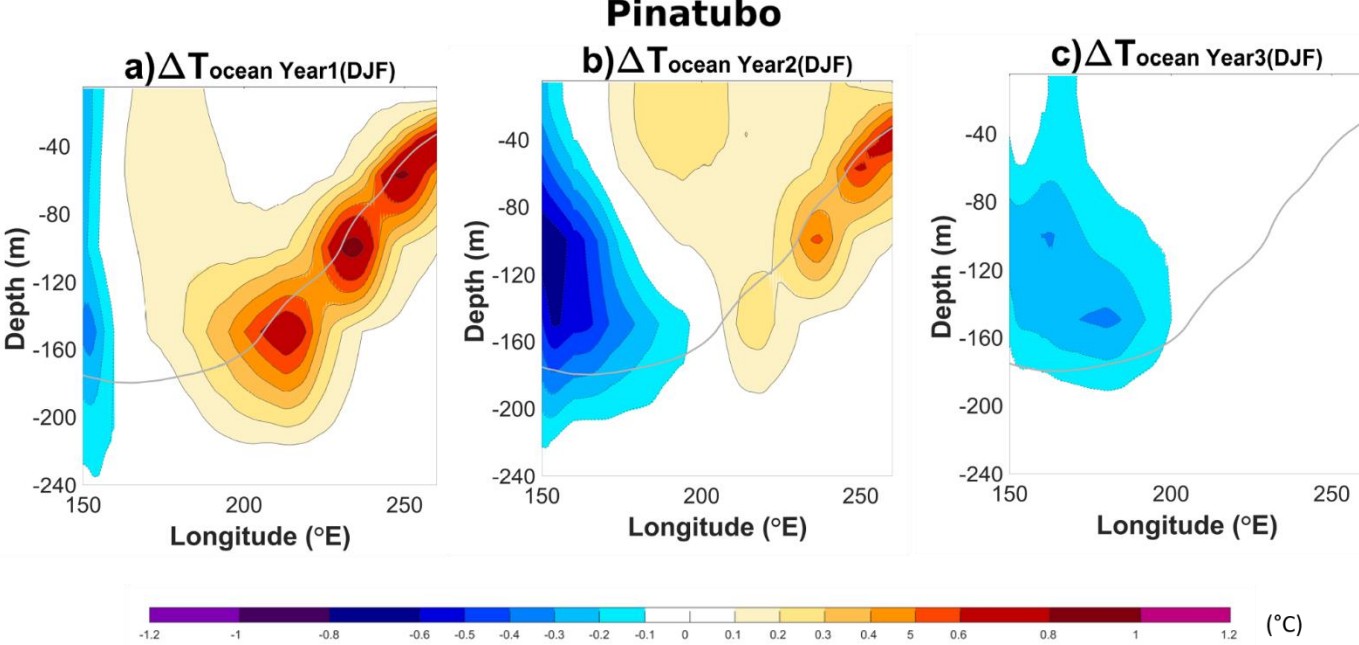

**Figure A16.** Ensemble mean changes shown for a transect in the equatorial Pacific (averaged 5°N – 5°S) of the ocean temperature (shadings) between the volcano case and the climatology for each of the following three winter season (DJF) after the Pinatubo eruption. Contours show the SST anomalies following the color bar scale (solid lines for positive anomalies and dashed lines lines for negative anomalies, the 0 line is omitted). The bold grey line shows the climatological thermocline depth (as defined using the 20°C isotherm). This is shown for 100 ensemble members.




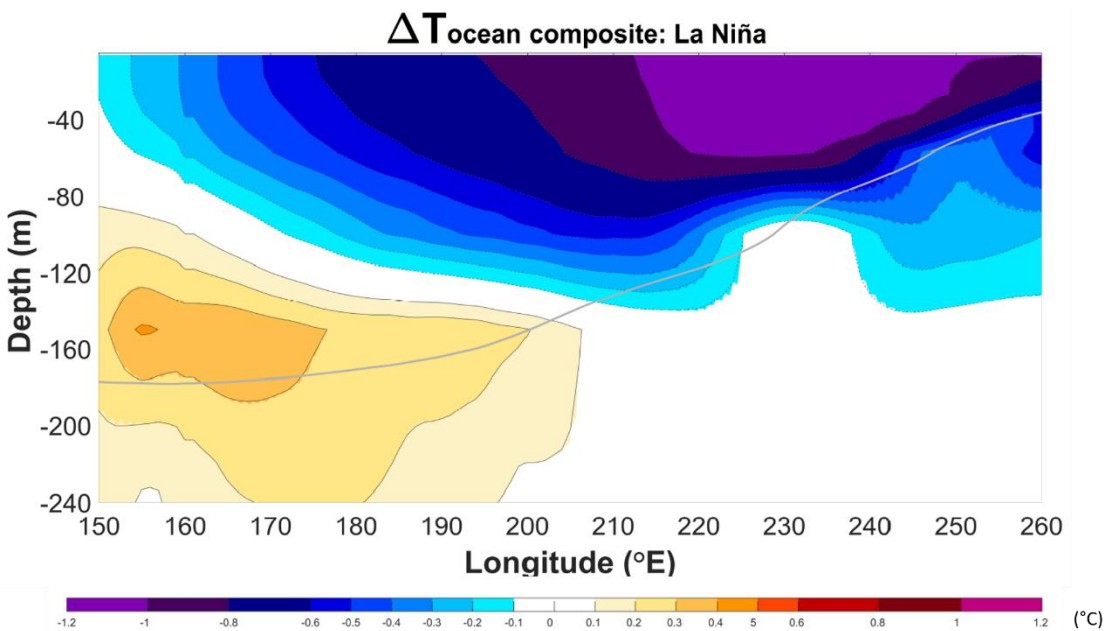

**Figure A17.** Temperature composite of La Niña events (Niño3.4 index < - 0.4°C) for a transect in the equatorial Pacific (averaged between 5°S and 5°N) in the reference period of each eruptions (4 years before each eruptions) and for the winter season (DJF). 100 ensemble members are considered, leading to a total of 1200 year as reference period and 237 La Niña events. Contours show the SST
anomalies following the colorbar scale (solid lines for positive anomalies and dashed lines for negative anomalies, the 0 line is omitted). The bold grey line shows the climatological thermocline depth (as defined using the 20°C isotherm).



**Model validation :**

Our model simulates a global cooling around 0.2-0.4°C following the 3 major eruptions investigated in the study (Agung, El Chichón and
Pinatubo). (Robock & Mao, 1995) provided an estimate for the cooling following these large volcanic eruptions to be of a similar magnitude
as those simulated by our model. Our model also reproduces the high-latitude winter warming after El Chichón and Pinatubo with
temperature anomalies exceeding locally 2°C.

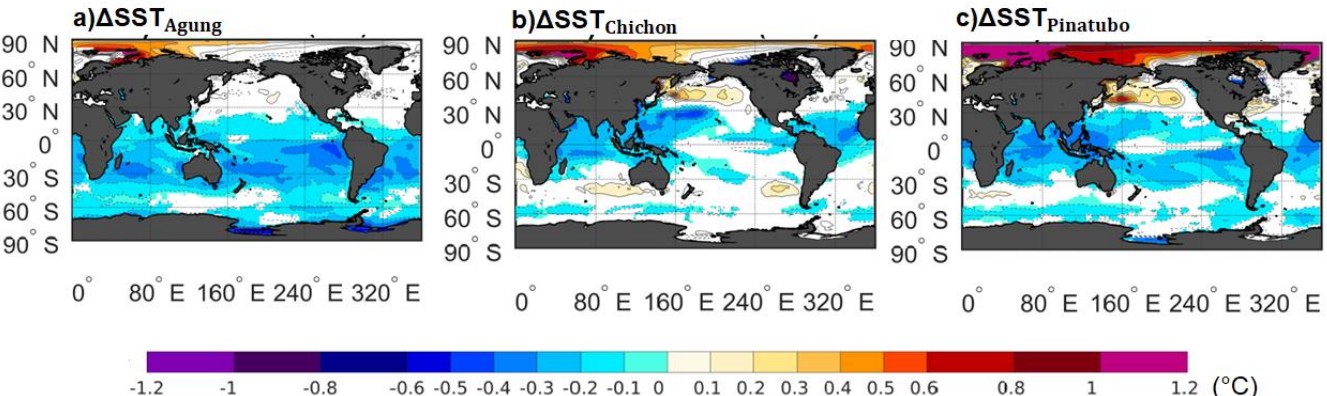

**Figure A18:** Ensemble mean of changes in sea surface temperature (SST) (shadings) between the volcano case and the climatology
for the year following the Agung (a-c), El Chichón (d-f) and the Pinatubo (g-i) eruptions. Contours show the SST anomalies following
the color bar scale (solid lines for positive anomalies and dashed lines lines for negative anomalies, the 0 line is omitted).

**Author contribution**

BW analysed the model output and wrote the manuscript with support of FSRP and NM. FSRP conceived the study and
supervised the findings of this work. NM provided the model output. All authors contributed to the interpretation of the results
and the writing of the manuscript.

**Competing interest**

The authors declare that they have no conflict of interest.

**Acknowledgements**

We thank Mikhail Dobrynin and Johanna Baehr from the University of Hamburg for completing the second hundred MPI-GE
ensemble simulations and providing the data from these simulations for use in this paper.

BW and FSRP acknowledge the financial support from the Natural Sciences and Engineering Research Council of Canada (grant RGPIN201804981) and the Fonds de recherche du Québec–Nature et technologies (2020NC268559). NM was supported by the Max Planck Society for the Advancement of Science and the Alexander von Humboldt Foundation.

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

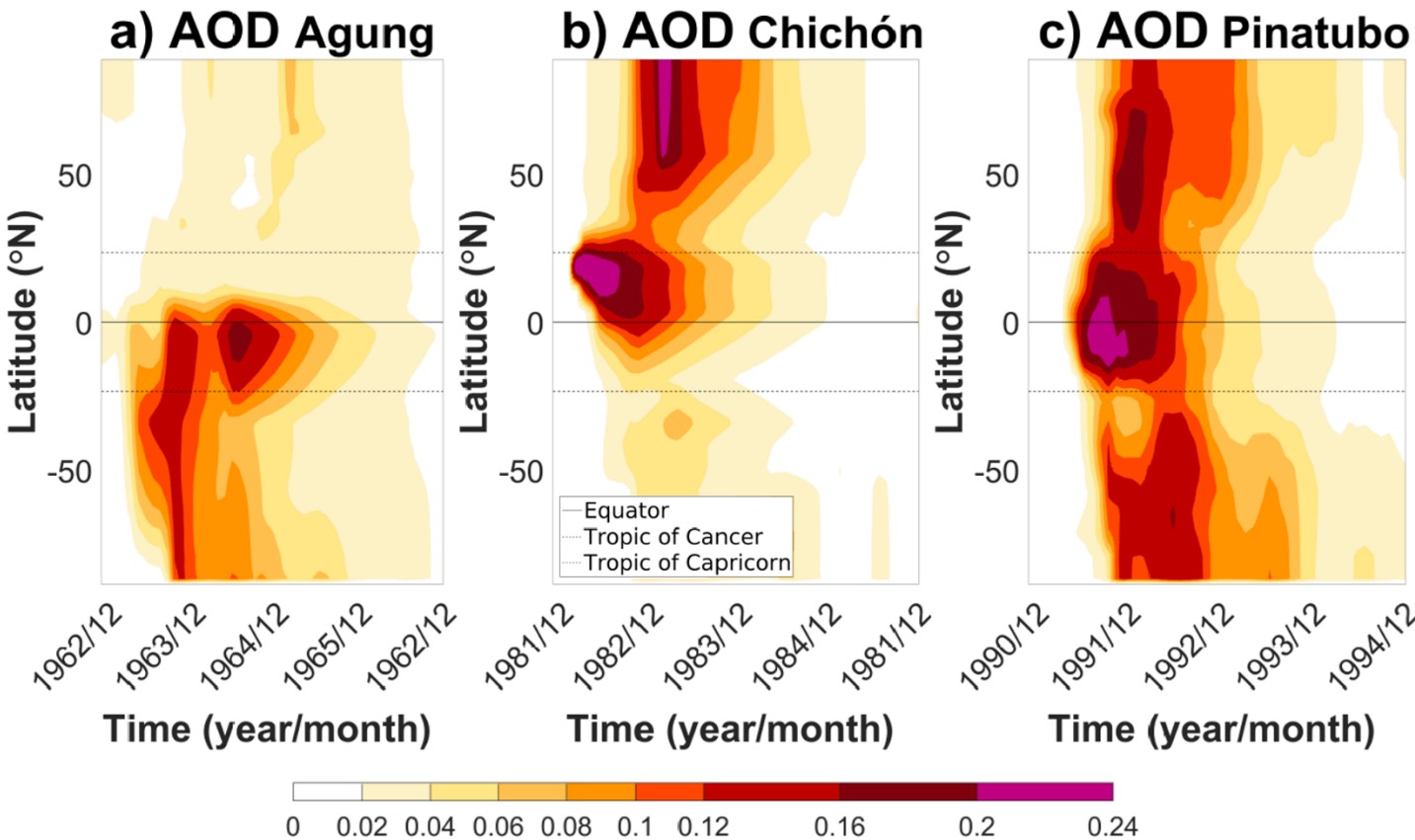

**Figure 1.** Evolution of the aerosol optical depth during four years for the three eruptions. The band of wavelength used is between approximatively 462 nm and 625 nm.


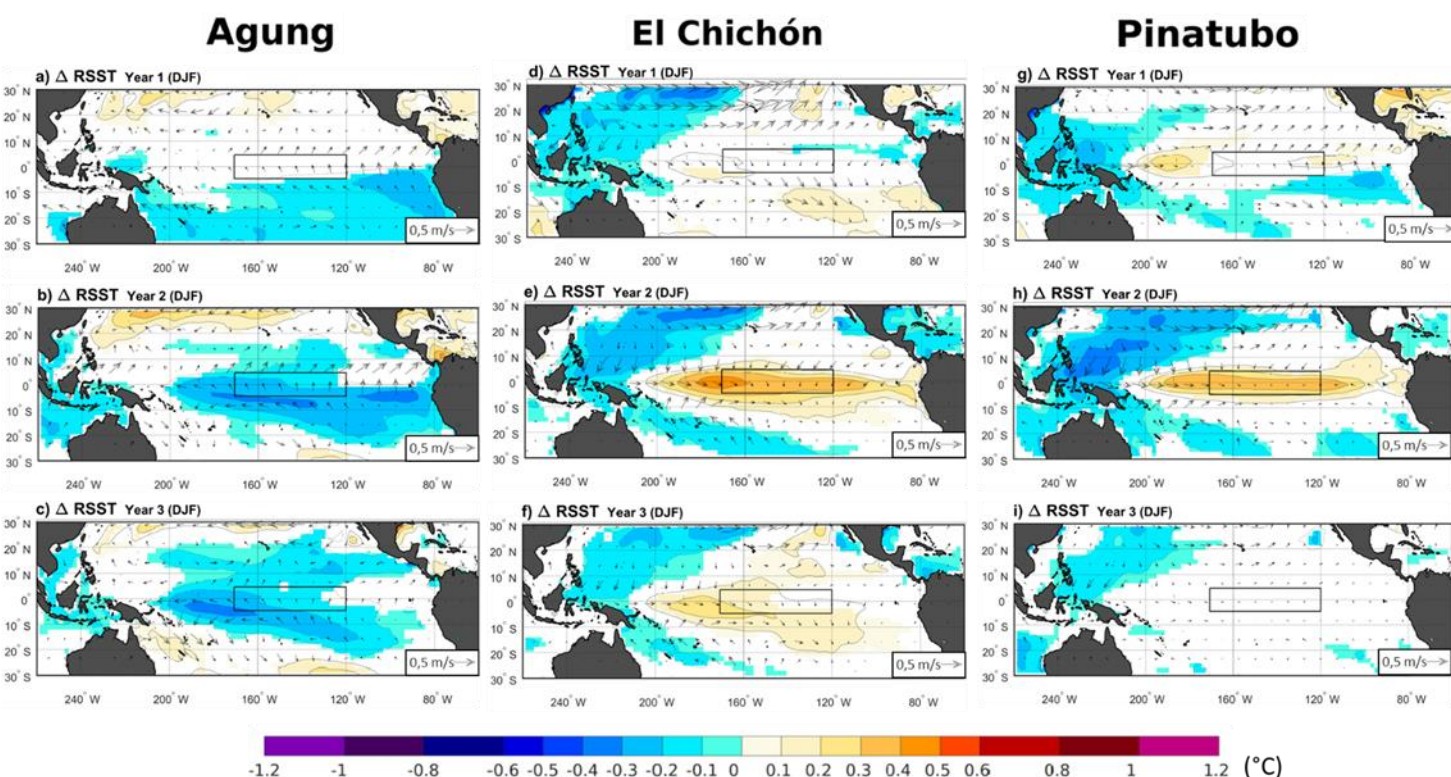

**Figure 2.** Ensemble mean of changes in relative sea surface temperature (RSST) (shadings) and 10 m winds (arrows) between the volcano case and the climatology for each of the following three winter season (DJF) after the Agung (a-c), El Chichón (d-f) and the Pinatubo (g-i) eruptions. Only significant RSST changes are shaded with an approximate 95 % confidence level using a Student t-test. Contours show the RSST anomalies following the color bar scale (solid lines for positive anomalies and dashed lines lines for negative anomalies, the 0 line is omitted). The boxes indicate the Niño 3.4 area.

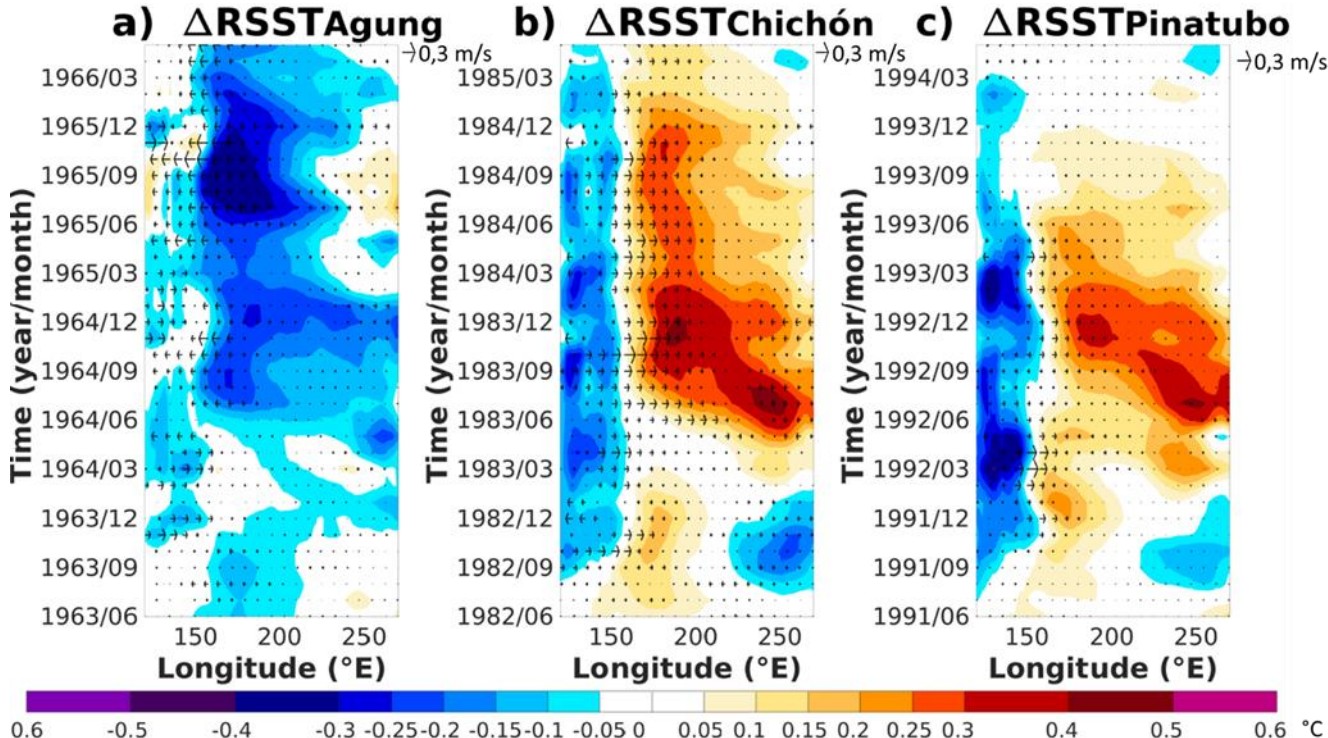

**Figure 3.** Hovmöller plot of the ensemble mean of the relative SST anomalies in the equatorial Pacific (averaged over -5 °N and 5 °N) and the change in the zonal component of the 10 m winds (m/s) for the three years following each eruption. The anomalies are calculated relative to the three years before each eruption.

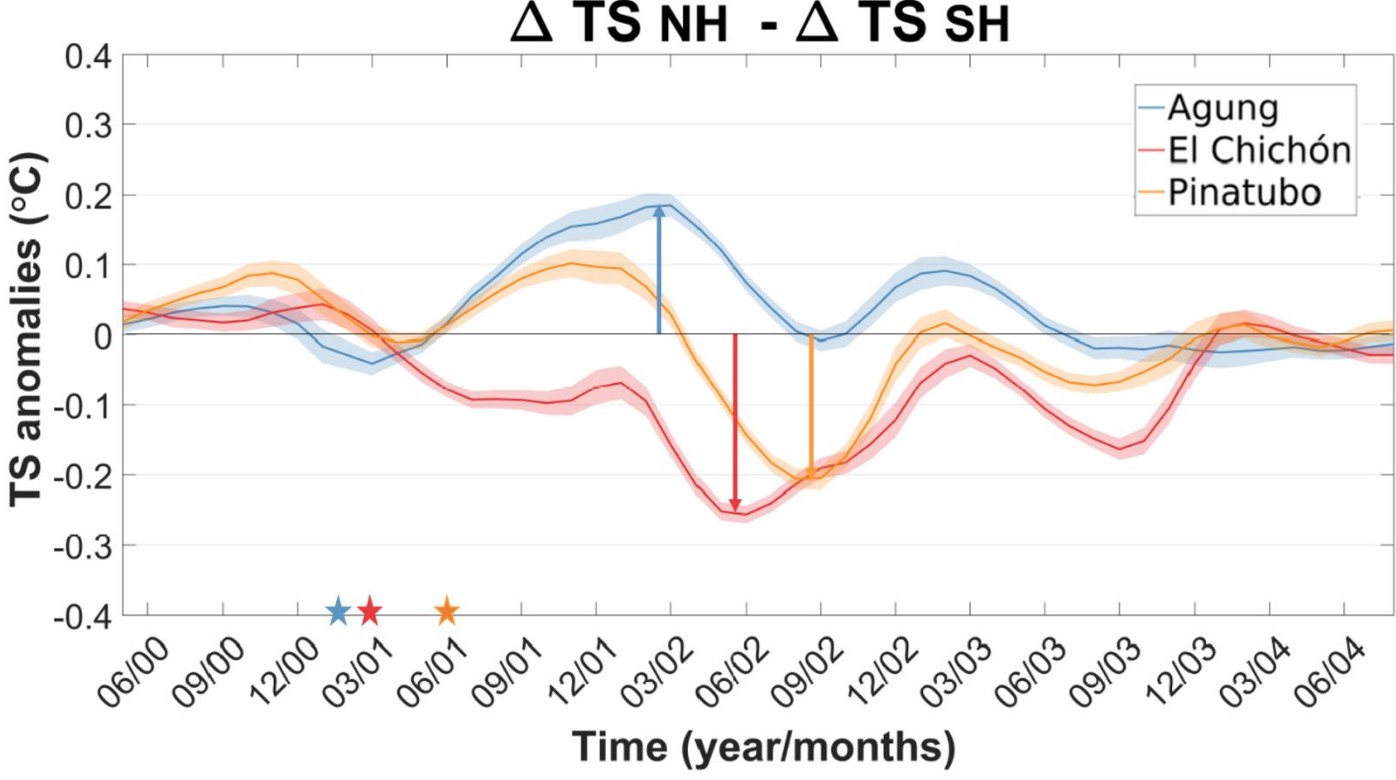

**Figure 4.** Evolution of the difference in the ensemble mean between the volcanically induced cooling of the SH and the NH after each eruption ($\Delta T_{NH} - \Delta T_{SH}$). Shading represents twice the standard error of the ensemble mean (i.e. ~95% confidence interval).

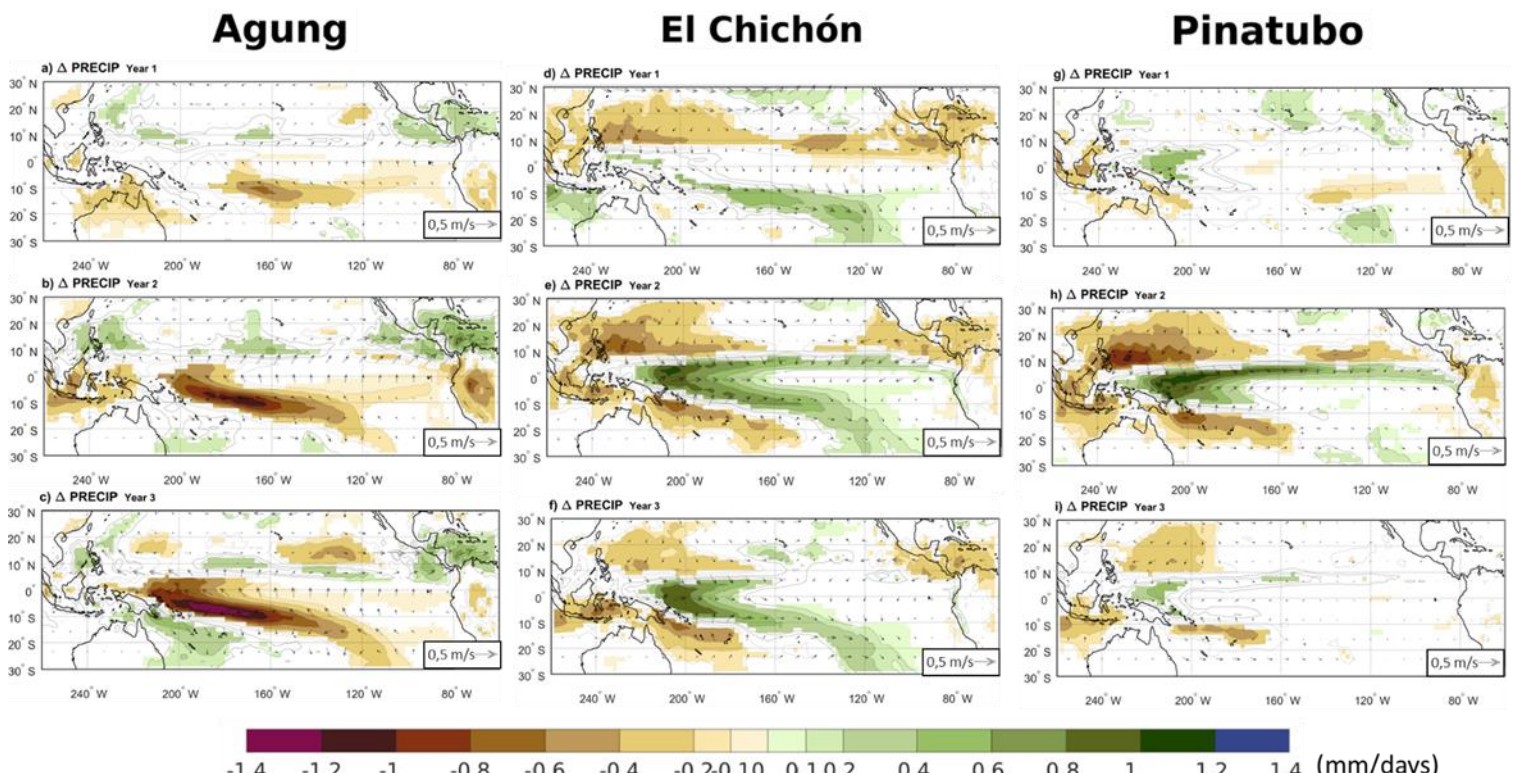

**Figure 5.** Change in ensemble mean precipitation and 10 m winds (arrows) between the references years and the volcano case for the three summer to winter seasons (June to February) following the Agung (a-c), El Chichón (d-f) and the Pinatubo (g-i) eruptions. Contours show the precipitation anomaly following the color bar scale (solid lines for positive anomalies and dashed lines for negative anomalies, the 0 line is omitted). Only precipitation changes that are significant at the 95% confidence level using a Student t-test are shaded.

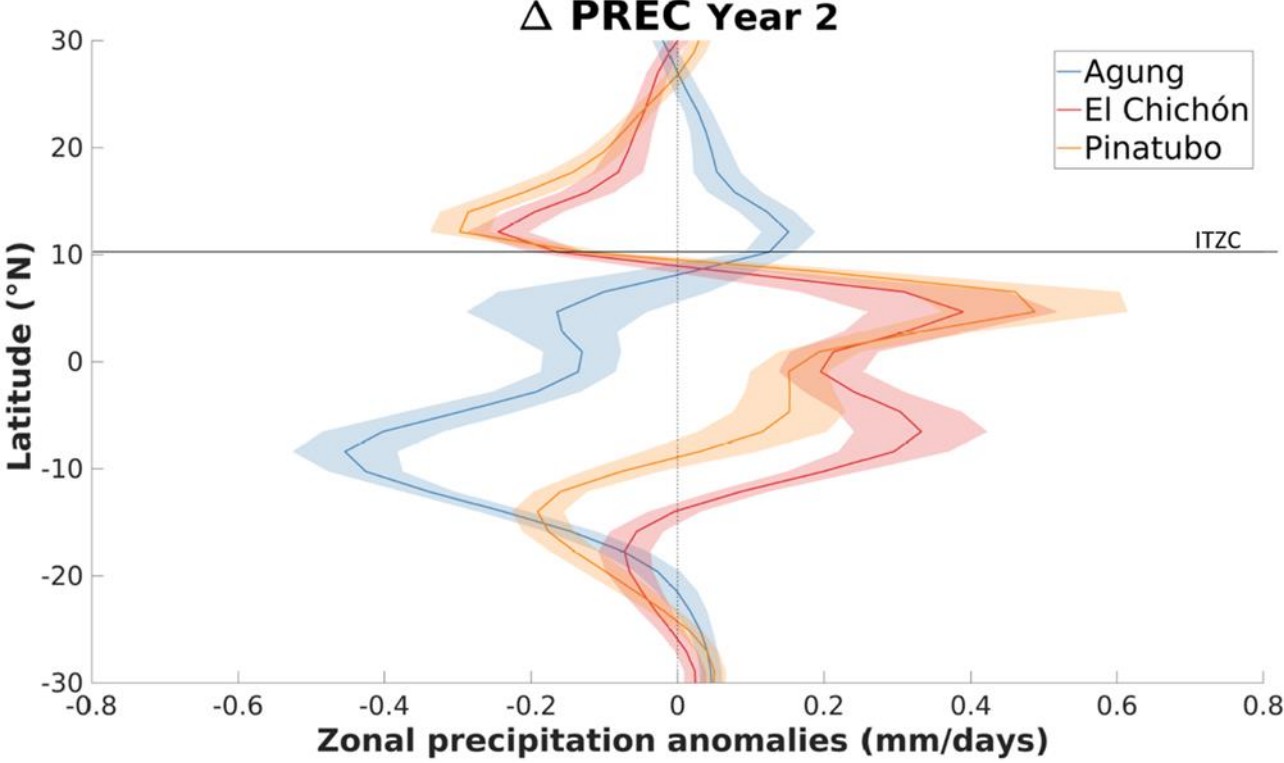


**Figure 6.** Ensemble mean of the zonal precipitation anomaly over the Pacific Ocean (160 °E-100 °W) between the summer to winter seasons (June to February) of the second year after each eruption and the 3-year climatology. Shading represents twice the standard error of the ensemble mean (i.e. 95% confidence interval). The horizontal line highlights the ensemble-mean 3-year climatology position of the ITCZ (defined as the location of the zonal-average precipitation maximum).

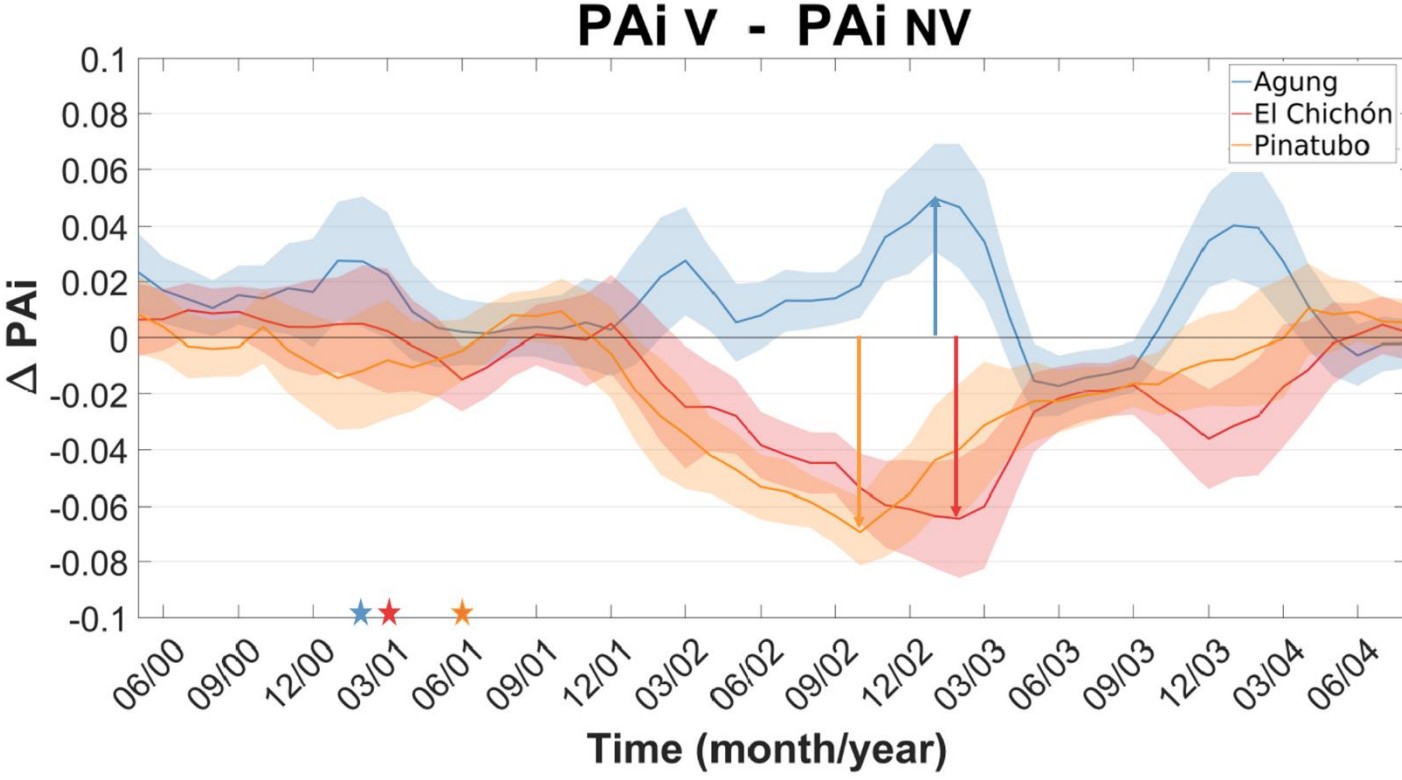

**Figure 7**. Evolution of the difference in the ensemble mean between the precipitation asymmetry index (Pai) after each eruption ($PAi_V$) and the climatology ($PAi_{NV}$). Shading represents twice the standard error of the ensemble mean (i.e. ~95% confidence interval). The three stars represents the moment of each eruption.