# Peer review of "The sensitivity of the ENSO to volcanic aerosol spatial distribution in the MPI Grand Ensemble"

_Earth System Dynamics, 2020_

## Referee Comment (RC1) · Anonymous Referee #1 · 12 Sep 2020

The reviewed paper deals with a very hot and scientifically important research question of how ENSO responds to hemispherically symmetric and asymmetric radiative forcing caused by the three largest volcanoes of the 20th century, Agung, El Chchon, and Pinatubo. The authors test the hypothesis that the primary driving mechanism is ITCZ's shift that works for symmetric and asymmetric volcanic forcing. The study takes advantage of the unique 200-member ensemble of MPI-ESM 1.1 with a relatively low effective spatial resolution, 180 km x 150 km x 16 layers. The research topic is important, the methods are sound, the results are interesting, but the analysis has significant flaws. Therefore I suggest the authors conduct a major revision to enhance their analysis and improve the text.

General concerns: The analysis is based on calculating the ensemble average re-

sponses to volcanic forcing. It is known that ENSO response to volcanic forcing is sensitive to the ocean preconditioning. The used approach averages out responses to individual initial conditions (El Nino, La Nino, or neutral), potentially removing the effect of the mechanisms that depend on preconditioning, e.g., ocean dynamical thermostat (ODT).

Nino3.4 index based on relative SST exaggerates the El Nino-like response. A comparative analysis with the regular SSt is needed.

Both EL Chihon and Agung eruption happened before the spring predictability barrier when ENSO responses are unstable. So, the responses to these two eruptions are contaminated by stochastic ENSO behavior.

ENSO's response to the Agung 1963 eruption, which is a milestone in the authors' arguments, is not what they think. It is not La Nina like a response, as its spatial temperature pattern dislikes the La Nina one. The ocean cooling in the Southern Hemisphere caused by Volcanic forcing that expands to the Northern Hemisphere could explain it. It is why it takes three years.

Specific Comments:

Fig. 1: Is it AOD from the model? Is it prescribed? From what data set? The color bar is wrong. AOD is an order of magnitude larger.

L36 and L89: The eruption started in February. The major emissions happened in March. Please use the month od the eruption consistently.

L39-44: RSST enhances the positive signal. Please show the regular SST results for comparison.

L66: Please specify that you are talking about Pacific ITCZ.

L70: Khodry et al. (2017) do not say anything about Walker Circulation. They claim that the atmospheric Kelvin wave does the job.
Interactive
comment

L121: Response on the 3rd year is too late even for ENSO responses driven by ocean Kelvin Wave propagation

L145-149: The response to the 1963 Agung eruption, as it is shown in Fig 2abc, is not a negative ENSO pattern. It is an ocean cooling but of different origin. It brakes the entire reasoning about the general applicability of the "ITCZ" mechanism.

L167: Why the reduction in precipitation should favor Rossby waves generation? Decreasing the release of latent heat should decrease the Rossby wave generation. Please clarify.

L185-186: For completeness of the analysis, it would be useful to have these experiments for this paper.

L210: In this particular model.

L225: Adams et al. saw the signal in two years after an eruption (year 0 and year 1).

---

## Referee Comment (RC2) · Anonymous Referee #2 · 18 Sep 2020

This study investigates the controversial topic of the influence of volcanic aerosol radiative forcing on ENSO. Model studies have suggested that volcanic forcing has a significant impact on tropical ocean-atmosphere circulation which manifest as anomalous ENSO signals. This study uses a very large ensemble of historical simulations, and explores the ENSO response to 3 large tropical eruptions, Agung, El Chichon, and Pinatubo, which each had contrasting spatial distributions of the aerosol forcing. The study shows a strong difference in ENSO response for the Agung eruption vs the other two eruptions, and links this difference in response to the differences in forcing structure. The authors then argue that this result supports a hypothesized mechanism linking volcanic aerosol forcing and ENSO, namely that hemispherically asymmetric forcing leads to a latitudinal shift in the ITCZ, leading to anomalous zonal winds at the

equator, and related changes in ocean current and SSTs.

The topic is certainly an active area of research that fits well in the scope of ESD. The model ensemble utilized is impressive in its size, which adds much to the statistical significance of results and aids the study to be able to contribute to the debate. However, the analysis does not sufficiently support the conclusions made, and I suggest the authors undertake major revisions before publication.

The main conclusion of the work is that the ENSO response is driven by the volcanically induced displacement of the ITCZ. It is not, I think, so clearly stated, but the implicit argument seems to be that the "other" proposed mechanisms such as ODT should depend only on the magnitude of the tropical forcing. The fact that different spatial patterns of forcing lead to (what appear to be) nearly opposite ENSO responses in the model results shown does seem to be a valid challenge to the ODT mechanism hypothesis. But, that in no way proves that the ITCZ mechanism is correct. This would only be the case if there were absolutely no chance of any other mechanism being important, and that is difficult to prove. There is not any strong evidence shown that the ITCZ shift actually happens, let alone that it is the cause of ENSO (or relative ENSO) anomalies. As a result, this main conclusion is not well supported by the results shown.

Specific comments:

Line 10 (and title): the paper focuses mostly on the *relative* ENSO, based on RSST anomalies. It is, I think, critically important to be clear with statements whether you refer to ENSO or relative ENSO. While the simulations show positive relative ENSO responses, the absolute SST anomalies seem to be small in this case, so, if ENSO is defined only by the Nino3.4 temperature index, these wouldn't qualify as positive ENSO. To put it another way, even a reader who is familiar with the topic would assume from this abstract that the model produces positive Nino 3.4 temperature anomalies from NH or symmetric forcings, which I believe is incorrect.

L12-13: "El Nino-like" or "La Nina-like" is used many times through the manuscript, without any description of what it means in comparison to the "actual" El Nino or La Nina. Why is it only "like" these things? How is it similar and/or different?

L13: again, I think you mean relative anomaly here?

L26: The atmospheric and ocean responses to volcanic forcing are difficult to prove from observations due to the limited sample size, and evidence for their existence is largely from models, which are imperfect. It's important not to overstate our confidence in the existence of these responses.

L19: I don't this Robock 2000 says this explicitly.

L31: Discussion of paleoclimate proxies should include the paper from Dee et al. (2020):

Dee, S. G., Cobb, K. M., Emile-Geay, J., Ault, T. R., Lawrence Edwards, R., Cheng, H. and Charles, C. D.: No consistent ENSO response to volcanic forcing over the last millennium, Science (80-. )., 367(6485), 1477–1481, doi:10.1126/science.aax2000, 2020.

L35: Please replace instances of "Krakatoa" with the preferred "Krakatau"

L39: Reference should be made in the paragraph to the analysis of CMIP5 models by Paik et al (2019): Paik, Seungmok, et al. "Volcanic-induced global monsoon drying modulated by diverse El Niño responses." Science Advances 6.21 (2020): eaba1212.

L40: "Most recent" does not necessarily mean most correct. Also, you say "studies" but cite only one paper.

L54: "due to the underlying dynamics of the Pacific Ocean" is vague and does not help the reader to understand this mechanism.

L66: Do you mean "moves southward"? Usually the ITCZ is fairly centered on the equator.

L66: One should give a precise definition of "trade winds". Also, please provide a clear explanation of how a meridional shift in the ITCZ produces a change in the magnitude of the trade winds, and not just a similar shift in their meridional location. This mechanism is central to the argument but is never given a physical basis.

L81: "volcano and no-volcano members" won't be clear to all readers.

L89: The size of eruptions refers to the amount of magma released. Here you are more interested in eruptions that produced the strongest stratospheric aerosol radiative forcing, which is not always the same as being the largest eruption.

L92: The aerosol loading from Pinatubo was *roughly* hemispherically symmetric, but not exactly.

L93: The actual hemispheric distribution for Krakatau is quite uncertain, since there are few observations from that time. The forcing used in the simulations is however roughly symmetric.

L103: The relevance of the "total of 600 years of climatology" is not clear.

L107: "over-estimate" rather than "amplify" would be more accurate.

L115: The source of the aerosol forcing distributions used in these simulations should be given, since it is very important when comparing the results shown here to other studies which use different forcing sets.

L117: I'm not sure we know for sure what processes are most important for controlling the hemispheric asymmetry of the aerosol forcing. Robock (2000) has argued that meteorological conditions at the time of the eruption are important, as it seems winds pushed the Pinatubo SO2 cloud to the equator. Otherwise, the Pinatubo distribution might have been more NH biased.

L125: Only 2 forcings are strongly asymmetric.

Fig 4: x axis label should be more specific (not "year"), and axis extended to include

the year of the eruption, with markers at the month of eruption.

L127: the description and Figure are at odds here, as the figure shows a difference in cooling between NH and SH, while the text mostly refers to hemispheric cooling.

L128: the magnitude of the eruptions is less important here than the magnitude of the forcing included in the simulations.

L131: This statement is not accurate, as strongest cooling seems to occur at the end of year 2.

L133: Unclear what "results" are referred to. Also, "are consistent with" is quite weak language–with the large amount of fields output by a climate model, one should be able to determine whether the Hadley Cell is displaced or not!

L134: There is nothing on Fig 5 that looks like a simple shift. On plots of anomalies, a shift should show up as a dipole, with negative and positive anomalies. The anomalies for Agung are almost only negative, while for EC and Pinatubo, the anomaly patterns look more like a narrowing of the ITCZ over the equator.

L136: This seems like a rather complicated hypothesis: simpler might be that the response is also sensitive to the magnitude of the forcing, which is stronger for Pinatubo?

L210: Nothing presented proves that these mechanisms are completely absent. They can still be active even if another process is more important to the overall response.

L211: A "limited number" of ensemble members is not a drawback per se: there are no studies that use an unlimited number of ensemble members.

L212: Why is reliance on statistical tools framed as a limitation of these prior studies?

---

## Author Comment (AC1) · 26 Jan 2021

The comment was uploaded in the form of a supplement:
https://esd.copernicus.org/preprints/esd-2020-63/esd-2020-63-AC1-supplement.pdf

———————————————————

---

## Author Response (AR1)

**Response to Reviewers**

We would like to thank both reviewers for their time and effort to review our manuscript. We have addressed all major and minor comments raised by the reviewers. In doing so, we feel we have crafted a revised manuscript that is more rigorous in content, and better presents the results of the study. Here we list the main changes that we have made:

- We now include an analysis of the SST, rather than only the RSST, adding 3 new figures, which are the equivalent of the figures 2, 3 and A2.
- We added a time series of the precipitation asymmetry index (PAi) to highlight the ITCZ shift.
- We added three supplementary figures to investigate the ocean temperature anomalies on a transect across the equatorial Pacific.

We did not investigate the impacts of the ocean initial conditions on ENSO, because this study aims to extend previous work that look at the response of single eruptions or a multiple-eruption mean to separate out the role of aerosol distribution of ENSO. The role of the initial conditions is indeed a relevant topic, which we have currently started investigating and will be the focus of a follow-up study.

Below, we copy the reviewers' comments in bold and describe how each of these issues has been addressed in the revised manuscript. The revised version of the manuscript is attached after the answers to reviewers' comments, and the changes compared to the original version are highlighted in bold.

**Response to Reviewer #1:**

**The reviewed paper deals with a very hot and scientifically important research question of how ENSO responds to hemispherically symmetric and asymmetric radiative forcing caused by the three largest volcanoes of the 20th century, Agung, El Chchon, and Pinatubo. The authors test the hypothesis that the primary driving mechanism is ITCZ's shift that works for symmetric and asymmetric volcanic forcing. The study takes advantage of the unique 200-member ensemble of MPI-ESM 1.1 with a relatively low effective spatial resolution, 180 km x 150 km x 16 layers. The research topic is important, the methods are sound, the results are interesting, but the analysis has significant flaws. Therefore, I suggest the authors conduct a major revision to enhance their analysis and improve the text.**

We thank the reviewer for the thorough evaluation of the manuscript and appreciate the positive remarks on the relevance of the study. We went through the analysis of the results (section 3 and 4) to make it clearer and we added the results with the SST and the ocean temperature transects to solidify our analysis.

**Major comments:**

**1. The analysis is based on calculating the ensemble average responses to volcanic forcing. It is known that ENSO response to volcanic forcing is sensitive to the ocean preconditioning. The used approach averages out responses to individual initial conditions (El Nino, La Nino, or neutral), potentially removing the effect of the mechanisms that depend on preconditioning, e.g., ocean dynamical thermostat (ODT).**

We agree with the reviewer that the initial conditions of the ocean before the eruptions have an impact on the ENSO response and that is a relevant aspect of the ENSO response to volcanic eruptions. However, our main object is to investigate the impact of different aerosols distributions (NH, SH and evenly distribution) rather than the impact of the initial conditions that is a matter of a follow-up study. Moreover, it is common to average the response of different initial conditions and investigating the averaged result when the objective is to present the response of ENSO to asymmetric forcing (e.g. Pausata et al., 2015; Stevenson et al., 2016; Zuo et al., 2018).

**2. Nino3.4 index based on relative SST exaggerates the El Nino-like response. A comparative analysis with the regular SST is needed.**

Following the reviewer suggestion, we added the analysis only looking at the SST (Fig A11, A12, A13). The results are really similar to the RSST except that the El Niño-like anomalies are, as expected, less strong since they are partly masked by the volcanically induced cooling. We have added the following comment in the manuscript comparing the RSST to the SST in the results section.

*"The results using the SST (Figs. A11, A12 and A13) are qualitatively similar to the RSST (Figs. 2, 3, and A3), but the El* Niño-*like anomalies are less strong since they are partially masked by the global cooling induced by the stratospheric aerosols which is in agreement with other studies (Khodri et al., 2017; Maher et al., 2015)."*

[Figure]

**Figure R1 :** Ensemble mean of changes in sea surface temperature (SST) (shadings) and 10 m winds (arrows) between the volcano case and the climatology for each of the following three winter season (DJF) after the Agung (a-c), El Chichón (d-f) and the Pinatubo (g-i) eruptions. Only significant SST changes are shaded with an approximate 95 % confidence level using a Student t-test. Contours show the SST anomalies following the color bar scale (solid lines for positive anomalies and dashed lines lines for negative anomalies, 60 the 0 line is omitted). The boxes indicate the Niño 3.4 area

[Figure]

**Figure R2:** Hovmöller plot of the ensemble mean of the SST anomalies in the equatorial Pacific (averaged over -5 °N and 5 °N) and the change in the zonal component of the 10 m winds (m/s) for the three years following each eruption. The anomalies are calculated relative to the three years before each eruption.

[Figure]

**Figure R3:** Ensemble mean changes in the Niño 3.4 index after each eruption. The 3-year climatology is subtracted to calculate the anomalies. Shading represents twice the standard error of the mean using an approximate 95% confidence interval.

**3. Both EL Chihon and Agung eruption happened before the spring predictability barrier when ENSO responses are unstable. So, the responses to these two eruptions are contaminated by stochastic ENSO behavior.**

While El Chichon and Agung eruptions did occur before the spring predictability barrier, the effect of the eruption and its associated aerosol loading is not limited to the time of the eruption. As it is possible to see from the AOD figure (Fig.1), the aerosol loading peaks in summer and high aerosol concentrations persist for a couple of years. In addition to that, given the large ensemble used in this study, the stochastic ENSO behaviour is filtered out (see also Milinski et al., 2019).

**4. ENSO's response to the Agung 1963 eruption, which is a milestone in the authors' arguments, is not what they think. It is not La Nina like a response, as its spatial temperature pattern dislikes the La Nina one. The ocean cooling in the Southern Hemisphere caused by Volcanic forcing that expands to the Northern Hemisphere could explain it. It is why it takes three years.**

The reviewer is not convinced that the spatial temperature pattern is La Niña-like, arguing that the cooling over the equatorial Pacific could just simply due to the direct cooling induced by the eruption. We thank the reviewer for raising this point as the analysis of the ocean transect along the equator across the tropical Pacific further strengthens our main conclusion (Figures A14, A15 and A16). We note that this analysis only includes 100-ensembles members due to data availability.

The temperature anomalies following Agung (Fig. A14) are not just superficial but extend down to 200 m depth. The warming below 100 m in the western Pacific is typical of an ongoing La Niña, as well as the cooling on the eastern side along the thermocline (increased upwelling). Furthermore, one can appreciate that the El Chichón and Pinatubo show indeed the opposite anomalies compared to Agung (Figs. A15 and A16). For example in the El Chichón case one can clearly appreciate the superficial cooling induced by the volcanic eruption on year 1, as a warming patter is simulated along the thermocline on the
eastern Pacific (typical of El Nino), but no warming at the surface is present.

The wind anomalies following Agung (see Fig. 2) are also opposite compared to El Chichón, hence leading to opposite upwelling conditions along the equator, further corroborating that the temperature anomalies are mostly dynamically driven. Therefore, this analysis strengthens our main conclusion on the role of the ITCZ in explaining the ENSO response to different aerosol distribution.

Furthermore, if the ENSO response were not dynamically driven (i.e. due to the induced changes in upwelling caused by wind anomalies), one would wonder why for Agung the aerosol loading over the equator does lead to a cooling extending deeper into the ocean, but a similar aerosol loading for El Chichón does not.

Finally, we are not sure how the ocean cooling in SH could expand to the NH (if it were not for dynamical reasons as the aerosol is confined to the SH). From figure R7, it is possible to see that the cooling "expanding" to the NH is just the La Niña
associated with the negative PDO pattern. To further strengthen the fact that those anomalies are indeed La Niña, we have performed a composite of all La Niña events occurring in the 4 years before each eruption (i.e. 12 years) for 100 ensemble members (total of 1200 years and 237 La Niña events). As shown in figure A17 the pattern is remarkably similar to what shown following the Agung eruption. Therefore, we can confidently assert that the Agung eruption triggered a La Niña response in our model simulation.

[Figure]

**Figure R4**: Ensemble mean changes shown for a transect in the equatorial Pacific (averaged 5°N – 5°S) of the ocean temperature (shadings) between the volcano case and the climatology for each of the following three winter season (DJF) after the Agung eruption. Contours show the SST anomalies following the color bar scale (solid lines for positive anomalies and dashed lines lines for negative anomalies, the 0 line is omitted). The bold grey line shows the climatological thermocline depth (as defined using the 20°C isotherm). This is shown for 100 ensemble members.

[Figure]

**Figure R5:** Ensemble mean changes shown for a transect in the equatorial Pacific (averaged 5°N – 5°S) of the ocean temperature (shadings) between the volcano case and the climatology for each of the following three winter season (DJF) after the Chichón eruption. Contours show the SST anomalies following the color bar scale (solid lines for positive anomalies and dashed lines for negative anomalies, the 0 line is omitted). The bold grey line shows the climatological thermocline depth (as defined using the 20°C isotherm). This is shown for 100 ensemble members.

[Figure]

**Figure R6:** Ensemble mean changes shown for a transect in the equatorial Pacific (averaged 5°N – 5°S) of the ocean temperature (shadings) between the volcano case and the climatology for each of the following three winter season (DJF) after the Pinatubo eruption. Contours show the SST anomalies following the color bar scale (solid lines for positive anomalies and dashed lines for negative anomalies, the 0 line is omitted). The bold grey line shows the climatological thermocline depth (as defined using the 20°C isotherm). This is shown for 100 ensemble members.

[Figure]

**Figure R7:** Ensemble mean of the changes in surface temperature between the climatology and the volcano case for each seasons of the year after Agung eruption. Only significant anomalies are shown with an approximate 95% confidence level using a Student *t*-test. Contours show temperature and precipitations anomalies following the color bar scale (solid line for positive anomalies and dashed line for negative anomalies).

[Figure]

**Figure R8:** Temperature composite of La Nina events (Nino3.4 index < - 0.4°C) for a transect in the equatorial Pacific (averaged between 5°S and 5°N) in the reference period of each eruptions (4 years before each eruptions) and for the winter season (DJF). 100 ensemble members are considered, leading to a total of 1200 year as reference period and 237 La Niña events. Contours show the SST anomalies following the colorbar scale (solid lines for positive anomalies and dashed lines for negative anomalies, the 0 line is omitted). The bold grey line shows the climatological thermocline depth (as defined using the 20°C isotherm).

**Minor Comments:**

**Fig. 1: Is it AOD from the model? Is it prescribed? From what data set? The color bar is wrong. AOD is an order of magnitude larger.**

We thank the reviewer for this useful comment, we were using a wrong spectral band for the AOD we fixed this problem in figure 1 and figure A6. In the methodology section we now read:

*"The stratospheric aerosols used in this study are prescribed in the historical simulations of the model (MPI ESM) from the data set of Stenchikov et al. (1998) (Giorgetta et al., 2013)."*

**L36 and L89: The eruption started in February. The major emissions happened in March. Please use the month of the**
**eruption consistently.**

Thank you for pointing that out, it has been changed in the revised manuscript.

**L39-44: RSST enhances the positive signal. Please show the regular SST results for comparison.**

We now included the SST (Figs. A11, A12 and A13) and we changed the terms La Niña-like and El Niño-like to relative La
Niña-like and relative El Niño-like throughout the manuscripts.

**L70: Khodry et al. (2017) do not say anything about Walker Circulation. They claim that the atmospheric Kelvin wave does the job.**

Thank you for pointing that out, Khodri et al. indeed refer to the fact that tropospheric *heating* anomalies in tropical Africa
generate anomalous atmospheric Rossby and Kelvin waves. Although they did not explicitly mention it, these anomalies consequently affect the Walker Circulation as they alter the trade winds and lead to an El Niño-like response (see Fig 7 in Khodri et al. (2017)). The Walker Circulation can be seen as a stationary Kelvin wave (see for ex. Stechmann & Ogrosky (2014)). It now reads:

*"Khodri et al. (2017) have invoked atmospheric teleconnections associated with the volcanically induced cooling of tropical Africa and weakening of the West African monsoon, which favours anomalous Rossby and Kelvin waves. These anomalous waves alter the Walker circulation and were identified as the primary cause of the post-eruption El Niño-like anomalies."*

**L121: Response on the 3rd year is too late even for ENSO responses driven by ocean Kelvin Wave propagation.**
We agree with the reviewer that the response on the 3[rd] year is too late for the ENSO response driven by ocean Kelvin Wave propagation. However, we are arguing that the ENSO response is driven by the ITCZ shift and not the Kelvin wave propagation due to the cooling of North Africa. Therefore, in Fig. 5 we see that there is still a strong northward shift of the ITCZ during the 3ʳᵈ years after the Agung and westward wind anomalies associated to this shift. For the Agung the global cooling last well into the third year and so does the asymmetric cooling of the hemispheres (Figs. A1 and 4).

**L145-149: The response to the 1963 Agung eruption, as it is shown in Fig 2abc, is not a negative ENSO pattern. It is an ocean cooling but of different origin. It brakes the entire reasoning about the general applicability of the "ITCZ" mechanism.**

As mentioned in the major issues we added in the appendix an ocean transect of the ensemble change in the ocean temperature (Fig. A14) that confirm that the ocean cooling after Agung is indeed due to a change in the ENSO state and not only a superficial cooling due to the presence of volcanic aerosols in the stratosphere. We also added a paragraph in the results:

*"Furthermore, the temperature anomalies following the Agung eruption are not just superficial but extend down to 200 m depth (Fig. A14). The warming below 100 m in the western Pacific and the cooling along the thermocline in eastern side of the basin are typical of an ongoing La Niña: the pattern is indeed remarkably similar to a composite of La Niña events occurring in the reference periods before the three eruptions (Fig. A17) . On the other hand, the temperature anomalies for the El Chichón and Pinatubo eruptions show opposite results as expected under El Niño conditions (Figs. A15 and A16). Moreover, the wind anomalies following Agung (see Fig. 2 and 3) are opposite compared to El Chichón, hence leading to opposite upwelling conditions along the equator, further corroborating that the ocean temperature anomalies are mostly dynamically driven through the Bjerknes feedback (Bjerknes, 1969)."*

**L167: Why the reduction in precipitation should favor Rossby waves generation? Decreasing the release of latent heat should decrease the Rossby wave generation.**

From Khodri et al.(2017) "…, it is hence the cooling over the largest tropical landmass, namely Africa, that has the strongest effect on Pacific winds (Fig. 5c). Indeed, the land surface cooling is maximum in the tropics during the first boreal summer and fall (Fig. 4j) and leads to a reduction of the West African monsoon (Fig. 6a, b). The reduced precipitation and tropospheric heating in the equatorial latitudes drive a Matsuno–Gill response[39, 40] where atmospheric equatorial Rossby and Kelvin waves induce easterly wind anomalies over the Atlantic and westerly wind anomalies over the Indian Ocean and western Pacific (Figs. 6a, b and 7 and Supplementary Fig. 11). This Kelvin wave suppresses convection along its path, with reduced convection over the western Pacific that further strengthens the westerly wind signal (Fig. 6a, b)."

We have modified the sentence to read:

*"More specifically, the reduction of the tropical precipitation and tropospheric cooling favours anomalous atmospheric Rossby and Kelvin waves in autumn (SON), with a weakening of the trade winds over the western Pacific, leading to El Niño-like conditions in the year after the eruption."*

**L185-186: For completeness of the analysis, it would be useful to have these experiments for this paper.**

While we agree with the review that such ad hoc sensitivity experiments will be useful, we do not have at the moment the computetional resources to run them, keeping in mind that we are talking about 200 ensemble simulations.

**L210: In this particular model.**

Thank you for pointing that out, it has been changed in the revised manuscript.

**L225: Adams et al. saw the signal in two years after an eruption (year 0 and year 1).**

Thank you for pointing that out. We have modified the sentences to read:

*"Furthermore, our model suggests the peak in ENSO anomalies to be on the second or third year after the eruption as in most modeling studies (Khodri et al., 2017; Lim et al., 2016; McGregor & Timmermann, 2011; Ohba et al., 2013; Stevenson et al., 2017), which is at odds with the reconstructions and observations that see a peak in ENSO anomalies in the first winter following the eruption (McGregor et al., 2010; 2020) and possibly extending to the second year (Adams et al., 2003). The delayed ENSO response in our model simulations relative to reconstructions and observations may be related to the apparent lack of extratropical-to-tropical teleconnections (Pausata et al., 2020) that could favour El Niño-like response already on the first winter following the eruption or other biases within the climate models (e.g., double ITCZ)."*

**Response to Reviewer #2:**

**This study investigates the controversial topic of the influence of volcanic aerosol radiative forcing on ENSO. Model studies have suggested that volcanic forcing has a significant impact on tropical ocean-atmosphere circulation which manifest as anomalous ENSO signals. This study uses a very large ensemble of historical simulations, and explores the**
**ENSO response to 3 large tropical eruptions, Agung, El Chichon, and Pinatubo, which each had contrasting spatial distributions of the aerosol forcing. The study shows a strong difference in ENSO response for the Agung eruption vs the other two eruptions, and links this difference in response to the differences in forcing structure. The authors then argue that this result supports a hypothesized mechanism linking volcanic aerosol forcing and ENSO, namely that hemispherically asymmetric forcing leads to a latitudinal shift in the ITCZ, leading to anomalous zonal winds at the**
**equator, and related changes in ocean current and SSTs.**

**The topic is certainly an active area of research that fits well in the scope of ESD. The model ensemble utilized is impressive in its size, which adds much to the statistical significance of results and aids the study to be able to contribute to the debate. However, the analysis does not sufficiently support the conclusions made, and I suggest the authors**
**undertake major revisions before publication.**

We thank the reviewer for the thorough evaluation of the manuscript and appreciate the positive remarks on the pertinence of the study. We went through the analysis of the results (section 3 and 4) to make it clearer and we added a time series of the precipitation asymmetry index (PAi) to support our conclusions.

**Major comments:**

**1. The main conclusion of the work is that the ENSO response is driven by the volcanically induced displacement of the ITCZ. It is not, I think, so clearly stated, but the implicit argument seems to be that the "other" proposed mechanisms such as ODT should depend only on the magnitude of the tropical forcing. The fact that different spatial patterns of forcing lead to (what appear to be) nearly opposite ENSO responses in the model results shown does seem to be a valid**
**challenge to the ODT mechanism hypothesis. But, that in no way proves that the ITCZ mechanism is correct. This would only be the case if there were absolutely no chance of any other mechanism being important, and that is difficult to prove. There is not any strong evidence shown that the ITCZ shift actually happens, let alone that it is the cause of ENSO (or relative ENSO) anomalies. As a result, this main conclusion is not well supported by the results shown.**

We agree with the reviewer that there may be additional mechanisms at play; however, the ones that are currently most used
(ODT, cooling of the Maritime Continent or of tropical Africa) are not supported by our model results for the reasons explained in the text (see L184 to L189 for ODT; see L190 to L197 for the Maritime Continent and see L198 to L207 for the tropical Africa cooling respectively). Among the possible mechanisms, only the ITCZ shift is fully supported by our results, in particular, the opposite results of El Chichón and Pinatubo vs. Agung. Moreover, our conclusions are that the ITCZ shift is the

*dominant* mechanism (not the exclusive one) that drives the post-eruption ENSO response because of the significant opposite response to the different volcanic forcing (NH, symmetrical and SH) over the 200 ensemble members. Nevertheless, we have toned down our conclusions and opened up for possible additional mechanisms, as we believe that more studies should indeed be done to investigate the ENSO response to volcanic eruptions.

Regarding the lack of evidence of ITCZ shift, we believed we have provided significant evidence for this. In particular: in figure 6 where it is possible to see the zonal rainfall anomalies and the opposite response for El Chichón and Pinatubo (with a significant increase of the precipitation south of the ITCZ and a significant decrease north) vs. Agung (with a significant increase of the precipitation north of the ITCZ and a significant decrease south); in figure 5 where the spatial distribution of the rainfall anomaly is shown, with a dipole pattern. It has been widely shown in the literature that ITCZ shifts occur as a direct thermodynamical response to differential/asymmetric cooling of the hemispheres (Kang et al., 2008; Schneider et al., 2014); this consequently do affect the trade winds and in turn, via the well-established Bjerknes feedback, the ENSO (Bjerknes, 1969).

Nevertheless, to further highlight the ITCZ shift and weakening we have performed additional analysis using the asymmetric precipitation index (Fig. 7).

[Figure]

**Figure R7**: Evolution of the difference in the ensemble mean between the precipitation asymmetry index (PAi) after each eruption ($PAi_V$). and the climatology ($PAi_{NV}$). Shading represents twice the standard error of the ensemble mean (i.e. ~95% confidence interval). The three stars represents the moment of each eruption.

In figure 7, we have calculated the precipitation asymmetry index in the same way of Colose et al. (2016) but around the average climatological mean of the ITCZ in our model, which is roughly 10 °N. So the expression is given by :

$$PAi = \frac{P_{20°N-10°N} - P_{10°N-Eq}}{P_{20°N-Eq}}$$

A positive PAi represents stronger precipitation north of the ITCZ and a negative PAi represents stronger precipitation south of the ITCZ. In figure 7, if the variation of the PAi is positive, it means that there is an augmentation of the precipitation north of the equator which indicates a northward shift of the ITCZ position. In the same way, if there is a negative variation of the PAi, there is a southward shift of the ITCZ. Our results confirm the opposite response in the ITCZ shift between Agung and El Chichón/Pinatubo. For Agung there is a significant northward shift of the ITCZ and for El Chichón and Pinatubo there is a significant southward shift of the ITCZ that peaks in the winter of the second year after the eruption (DJF year2).The inclusion of this figure in the main results support our initial claim that there is a ITCZ shift associated with these eruptions. In the results section we now read:

*"Moreover, the precipitation asymmetry index (PAi) (Colose et al., 2016) further highlight the ITCZ shift and weakening (Fig. 7). The expression for the PAi is given by:*

$$PAi = \frac{P_{20°N-10°N} - P_{10°N-Eq}}{P_{20°N-Eq}}$$

*It represents the variation of the zonal precipitation around the average climatological mean of the ITCZ in our model, which is roughly 10 °N. In figure 7, the positive variation of the PAi after Agung eruption means a northward shift of the ITCZ and the negative variations of the PAi after Pinatubo and El Chichón eruptions means a southward shift of the ITCZ. "*

**Minor comments:**
**L10 (and title): the paper focuses mostly on the \*relative\* ENSO, based on RSST anomalies. It is, I think, critically important to be clear with statements whether you refer to ENSO or relative ENSO. While the simulations show positive relative ENSO responses, the absolute SST anomalies seem to be small in this case, so, if ENSO is defined only by the Nino3.4 temperature index, these wouldn't qualify as positive ENSO. To put it another way, even a reader who is familiar with the topic would assume from this abstract that the model produces positive Nino 3.4 temperature anomalies from NH or symmetric forcings, which I believe is incorrect.**

We agree with the reviewer and we replaced ENSO response to relative ENSO response in the manuscript. We also included
the analysis with the SST and the ocean temperature transects in the appendix to show that the RSST results are similar than the SST results but just partially masked by the volcanic cooling and that the anomalies are dynamically driven by an alteration of the ENSO state.

**L12-13: "El Nino-like" or "La Nina-like" is used many times through the manuscript, without any description of what**
**it means in comparison to the "actual" El Nino or La Nina. Why is it only "like" these things? How is it similar and/or different?**

We thank the reviewer for pointing that out as "El/La Niño/a-like" has been widely used in the literature with different meanings (in particular model vs. proxy community). We have added a new sentence in the introduction (section 1) of the revised manuscript to clarify this point:

"*The terms El* Niño-*like and* La Niña-*like conditions are used in the context of modeling studies to describe an anomalous warming or cooling of the equatorial Pacific relative to the reference case.*"

**L13: again, I think you mean relative anomaly here?**
We thank the reviewer for pointing that out, we are now referring to relative anomalies throughout the manuscript.

**L26: The atmospheric and ocean responses to volcanic forcing are difficult to prove from observations due to the limited sample size, and evidence for their existence is largely from models, which are imperfect. It's important not to overstate our confidence in the existence of these responses.**
We have rephrased the sentence to read:

"*These rapid modifications in temperature may induce dynamical changes in the atmosphere and in the ocean such as a strengthening of the polar vortex ...*"

**L19: I don't this Robock 2000 says this explicitly.**
We have changed the sentence to read:

"*Aerosol particles from volcanic eruptions are one of the most important non-anthropogenic radiative forcing that have influenced the climate system in the past centuries (e.g., Robock, 2000).*"

**L31: Discussion of paleoclimate proxies should include the paper from Dee et al.**
**(2020):**

**Dee, S. G., Cobb, K. M., Emile-Geay, J., Ault, T. R., Lawrence Edwards, R., Cheng, H. and Charles, C. D.: No consistent ENSO response to volcanic forcing over the last millennium, Science (80-. )., 367(6485), 1477–1481, doi:10.1126/science.aax2000, 2020.**

We thank the reviewer to bring this new paper that add more debate on this controversial subject. We now read in the introduction :

*"Paleoclimate archives and observations from the past centuries suggested that large tropical eruptions are usually followed by a warm sea-surface temperature (SST) anomaly in the Pacific (e.g., Adams et al. , 2003; D'Arrigo et al. , 2005; Li et al., 2013; S. McGregor et al. , 2010; Wilson et al., 2010) even if there is still uncertainty on the significance of theses results (Dee et al., 2020)."*

**L39: Reference should be made in the paragraph to the analysis of CMIP5 models by Paik et al (2019): Paik, Seungmok, et al. "Volcanic-induced global monsoon drying modulated by diverse El Niño responses." Science Advances 6.21 (2020): eaba1212.**

We thank the reviewer for bringing this recent paper to the discussion.

**L35: Please replace instances of "Krakatoa" with the preferred "Krakatau"**

We changed the occurrence of "Krakatoa" to "Krakatau"

**L40: "Most recent" does not necessarily mean most correct. Also, you say "studies" but cite only one paper.**

We agree with the reviewer that most recent studies does not necessarily mean most correct we changed the sentence so we now read:

*"However, the majority of the studies have pointed to an El Niño-like response following volcanic eruption (for a review of these studies see McGregor et al., 2020)."*

(McGregor et al., 2020) is a review paper on our understanding of the impact the strong volcanic eruption on ENSO, so the paper itself refers to numerous other studies.

**L54: "due to the underlying dynamics of the Pacific Ocean" is vague and does not help the reader to understand this** 370 **mechanism.**

We clarified the ocean dynamical thermostat mechanism to make it clearer for the reader:

*"due to the preferential cooling of the warm pool in the western Pacific in comparison to the upwelled water in the eastern Pacific."*

**L66: Do you mean "moves southward"? Usually the ITCZ is fairly centered on the equator.**

To avoid confusion, we have changed it to read "southward"

**L66: One should give a precise definition of "trade winds". Also, please provide a clear explanation of how a meridional**
**shift in the ITCZ produces a change in the magnitude of the trade winds, and not just a similar shift in their meridional location. This mechanism is central to the argument but is never given a physical basis.**

We agree that since the trade winds are a central argument in the ITCZ mechanism this concept require more attention.
We added to the introduction:

*"The mechanisms that trigger a change in the ENSO state following volcanic eruptions are still debated. The center of the argument is explaining how a volcanic eruption weaken or amplify the trade winds (i.e. constant surface easterly winds within the tropics)."*

Regarding how the ITCZ shift produces a change in the magnitude of the trade winds we have added the following sentence
in the manuscript:

*"Trade winds converge towards the ITCZ where their intensity weakens, as the air circulates in an upward direction, creating the so-called doldrums (areas of windless waters). Consequently, a shift of the ITCZ affects the position of the doldrums and the intensity of the trade winds over the equator.*

**L81: "volcano and no-volcano members" won't be clear to all readers.**

We thank the reviewer for bringing this point, why clarified the definition of volcano and no-volcano members:

*"Other studies use a small number of ensemble members with volcanic forcing and ensemble members without volcanic forcing*
*starting from different initial conditions …"*

**L89: The size of eruptions refers to the amount of magma released. Here you are more interested in eruptions that produced the strongest stratospheric aerosol radiative forcing, which is not always the same as being the largest eruption.**
We thank the reviewer for pointing this out we now read:

*"In this study we consider the three largest eruptions, in terms of quantity of aerosols injected in the stratosphere, of the last 100 years; the Agung in Indonesia (8 °S 115 °E) in February 1963; El Chichón in Mexico (17 °N 93 °W) in March 1982; and the Pinatubo in the Philippines (15 °N 120 °E) in June 1991."*

**L92: The aerosol loading from Pinatubo was \*roughly\* hemispherically symmetric, but not exactly**

We thank the reviewer for pointing this out we changed the phrasing.

**L93: The actual hemispheric distribution for Krakatau is quite uncertain, since there are few observations from that**
**time. The forcing used in the simulations is however roughly symmetric.**

We thank the reviewer for pointing this out we added this nuance.

**L103: The relevance of the "total of 600 years of climatology" is not clear.**

Thank you for pointing that out we clarified this point in section 2, we now read:

*"The anomalies calculated are the difference between the reference (3 years before the eruption and for the 200 ensembles members that we refer to as climatology) and periods after the eruption."*

**L107: "over-estimate" rather than "amplify" would be more accurate.**
Thank you for pointing that out.

**L115: The source of the aerosol forcing distributions used in these simulations should be given, since it is very important when comparing the results shown here to other studies which use different forcing sets.**

We thank the reviewer for this pertinent remark we added in the methodology:

*"The stratospheric aerosols used in this study are prescribed in the historical simulations of the model (MPI ESM) and comes from the data set of Stenchikov et al. (1998) (Giorgetta et al., 2013)."*

**L117: I'm not sure we know for sure what processes are most important for controlling the hemispheric asymmetry of**
**the aerosol forcing. Robock (2000) has argued that meteorological conditions at the time of the eruption are important, as it seems winds pushed the Pinatubo SO2 cloud to the equator. Otherwise, the Pinatubo distribution might have been more NH biased.**

Thank you for this remark, we added nuance to the statement.

**L125: Only 2 forcings are strongly asymmetric.**

Thank you for pointing that out we changed the sentence to :

*"The different forcing caused by the three eruptions induces a different cooling of the surface temperature in the two hemispheres (Fig. 4)."*

**Fig 4: x axis label should be more specific (not "year"), and axis extended to include the year of the eruption, with markers at the month of eruption.**

We agree that the axis should be extended to see the years of the eruptions. We modified the figure.

**L127: the description and Figure are at odds here, as the figure shows a difference in cooling between NH and SH, while the text mostly refers to hemispheric cooling.**

We thank the reviewer for pointing out this inconsistency. What we wanted to say is that if there is a difference of cooling between the NH and the SH it's because one of the two hemispheres has been more cooled than the other and thus, there is a hemispheric cooling of an hemisphere relative to the other hemisphere. To clarify this point we added relative hemispheric cooling in the discussion.

**L128: the magnitude of the eruptions is less important here than the magnitude of the forcing included in the simulations.**

We agree with the reviewer that the amplitude of the eruption in terms of magma ejected is less important than the quantity of stratospheric aerosols. We now read in the results section:

*"The relative hemispherical cooling associated to Agung (SH) is in absolute values comparable to Pinatubo even if Agung's quantity of aerosols injected in the stratosphere was twice as small (Bluth et al., 1992; Self & Rampino, 2012)"*

**L131: This statement is not accurate, as strongest cooling seems to occur at the end of year 2.**

We agree that the cooling is not exactly peaking at the beginning of year 2, but it is also not peaking at the end of the year since El Chichón and Pinatubo peak in June. The point is that the maximum cooling and the difference of cooling between the two-hemispheres peak roughly in the same time. We changed the text and we now read:

*"The maximum cooling for all three eruptions occurs in the second year and so does the temperature difference between the two hemispheres".*

**L133: Unclear what "results" are referred to. Also, "are consistent with" is quite weak language–with the large amount of fields output by a climate model, one should be able to determine whether the Hadley Cell is displaced or not!**

We rephrased the sentence with a stronger affirmation:

*Precipitation anomalies (Figs. 5 and 6) show the displacement of the Hadley cell associated with the differential cooling between the hemispheres, showing a northward shift of the ITCZ for the eruption of the Agung, and a southward shift for both El Chichón and the Pinatubo, for the three years following the eruption.*

**L134: There is nothing on Fig 5 that looks like a simple shift. On plots of anomalies, a shift should show up as a dipole, with negative and positive anomalies. The anomalies for Agung are almost only negative, while for EC and Pinatubo, the anomaly patterns look more like a narrowing of the ITCZ over the equator.**

The dipole pattern is present on figure 5 for the three eruptions: For Agung we can see a small but significant increase of the precipitation north of the ITCZ and a strong reduction of the precipitation south of the ITCZ. Moreover, we can also see the westward winds anomalies associated with the reduction of precipitation, weakening the trade winds and triggering a La Niña-like anomaly via Bjerknes feedback. There is also a similar opposite dipole pattern for El Chichón and Pinatubo in comparison to Agung and it is the main reason why we argue that the ITCZ shift is a dominant mechanism in the ENSO response. Figure 7 also supports that there is a significant northward shift of the ITCZ after Agung and a significant southward shift after El

Chichón and Pinatubo.

**L136: This seems like a rather complicated hypothesis: simpler might be that the response is also sensitive to the magnitude of the forcing, which is stronger for Pinatubo?**

We thank the reviewer for the suggestion. We have included it in the text along with the global warming recently proposed hypothesis (Fasullo et al., 2017) that we believe is worth further investigation.

*"This could indicate that the ITCZ response is sensitive to the magnitude and spatial distribution of the forcing. The on-going global warming could also amplify the rainfall anomaly following the volcanic eruptions through modulation in the ocean stratification and near-surface winds amplifying the response as suggested in a recent study (Fasullo et al., 2017) (Fig. A9)."*

**L210: Nothing presented proves that these mechanisms are completely absent. They can still be active even if another process is more important to the overall response.**

We agree with the reviewer, we changed the sentence so we clearly say that it is not that these mechanisms are absent it is just that we can not say that they are dominant since we observe a different response for symmetrical and NH in comparison to SH

aerosol distribution. It now reads :

*"Our work also pointed out that the ODT (Clement et al., 1996), the cooling of the Maritime Continent (Ohba et al., 2013) and of the tropical Africa (Khodri et al., 2017) mechanisms are not dominant for the ENSO response following volcanic eruptions in our model."*

**L211: A "limited number" of ensemble members is not a drawback per se: there are no studies that use an unlimited number of ensemble members.**

We thank the reviewer for pointing that out, we changed the occurrence of limited to small throughout the manuscript.

**L212: Why is reliance on statistical tools framed as a limitation of these prior studies?**

The reliance on statistical tools is a limitation as it is based on a small sample of events: for example, Liu et al. (2018) use a synthesis of publicly available proxy-based ENSO reconstructions and two 1500-year long simulations with and without volcanic forcing (no ensemble members). They perform the superposed epoch analysis (SEA) on both proxy data and volcano experiment, concluding that the NH and tropical eruptions lead to El Niño-like conditions in the year following the eruption, while SH eruptions produce a different behavior (La Niña for the proxies; very weak response for the model simulation). The SEA is often used to test whether the El Nino following the eruption is significant or not, using proxy data. However, their SEA analysis show that even before the eruptions there are quite significant NINO3 responses for all eruptions (NH, SH and tropical). It is difficult to determine whether the NINO3 anomalies following the eruptions are induced by the eruptions and hence why we need large ensemble to be able to pull apart the stochastic variability from the volcano signal. To clarify this point we have modified that sentence in the discussion :

[revised manuscript text omitted]

---

## Author Response (AR2)

**Response to Reviewers**

We would like to thank both reviewers for their time and effort taken to provide this second review our manuscript. We have addressed all major and minor comments raised by the reviewers. In doing so, we feel we have crafted a revised manuscript that is more rigorous and better includes the most recent studies on the subject. Here we list the main changes that we have made:

- We added a composite analysis when the ITCZ is north or south of its climatological location in order to investigate the wind anomaly along the equator and compare it with the volcano cases.

- We further toned down our discussion and conclusion, and highlight the caveat associated to our analysis since we do not investigate the role of the initial conditions, which may be important in affecting the ODT mechanism.

Below, we copy the reviewers' comments in bold and describe how each of these issues has been addressed in the revised manuscript and the line in the revised manuscript (RM) are identified. The RM is attached after the answers to reviewers' and the changes compared to the previous version are highlighted in bold.

**Response to Reviewer #1:**

**This is the second review, so I skip the general comments on the paper content. I appreciate the author's efforts in improving the paper, but serious discrepancies remain. I also have to mention that despite the real volcanoes are chosen for the analysis, there are no attempts to compare the model results with observations.**

We thank the reviewer for this second evaluation of the manuscript and appreciate the positive remarks. Our model simulates a global cooling of around 0.2°C following the 3 major eruptions investigated in the study (Agung, El Chichón and Pinatubo). Mau and Robock, 1995 provided an estimate for the cooling following these large volcanic eruptions to be of a similar magnitude as those simulated by our model. Our model also reproduces the high-latitude winter warming after El Chichón and Pinatubo in the first winter after the eruption locally 2°C.

[Figure]

**Figure R1:** Ensemble mean of changes in sea surface temperature (SST) (shadings) between the volcano case and the climatology for the year following the Agung (a-c), El Chichón (d-f) and the Pinatubo (g-i) eruptions. Contours show the SST anomalies following the color bar scale (solid lines for positive anomalies and dashed lines lines for negative anomalies, the 0 line is omitted). MPI-ESM has been extensively used to look at the impacts of volcanic eruptions on climate (e.g. Bitner et al. 2016, Timmreck et al, 2016, Illing et al., 2018). In particular, Bittner et al., 2016 validate the zonal mean zonal wind anomalies between 55°N – 65°N at 10 hPa (m/s) in the first winter after eruptions in the Supplementary Material, showing that the zonal mean temperature and wind anomalies in DJF are fairly realistic in the model. Bittner also wrote a more extensive thesis where he compares some observed fields to the model version used in our study (Bittner 2015).

We added figure R1 in the appendix as model validation and we added the following paragraph in the methodology section (line 118 to 123 in the RM):

*"MPI-ESM has been extensively used to investigate the impacts of volcanic eruptions on climate (e.g. Bittner et al., 2016; Illing et al., 2018; Timmreck et al., 2016). In particular our model is able to reproduce the global cooling of around 0.2 following the three eruptions investigated in this study as well as the high-latitude winter warming (locally up to 2°C) after El Chichón and Pinatubo in the first winter after the eruption (see Fig. A18) in agreement with the estimates provided in the literature (e.g. Robock & Mao, 1995; Timmreck, 2012)."*

**Minor Comments:**

**R1. It has been reported by Predybaylo et al. (2017, 2020) that the ODT effect is sensitive to ocean preconditioning. You can not claim that ODT is not working if you did not test it properly. Please remove the claim that you keep in the revised paper or make a proper analysis.**

We agree with the Reviewer that to claim that the ODT mechanism is not working, a specific set of sensitivity experiments need to be performed. Here we like to clarify that we do not think that the ODT *effect* is not present. The effect is present in both Pinatubo and El Chichón as the ENSO responds to a global cooling with a positive phase (warming), which means the ODT is indeed at play.

However, what we are claiming is that the ODT may not be the main *mechanism* in our model, but rather a *response* as also explained in Pausata et al. (2020). The studies that the Reviewer cited did not explicitly test the ODT mechanism but rather the response, nevertheless they claimed the existence of an ODT *mechanism*. Predybaylo et al., 2017 used a diagnostic tool to assess the *presence* of the ODT and main contributors, but as it is a diagnostic tool, it cannot be used to determine whether the ODT is a mechanism or rather a response to other mechanisms, triggering an ODT response. Predybaylo et al., 2020 performed a set of sensitivity experiments in which they prescribed land temperature to attribute the responses to volcanically induced land-ocean temperature gradients and the equatorial Pacific Ocean-dynamical thermostat. However, by imposing the cooling of land (Maritime Continent + Australia) they are assuming the response of the Equatorial Pacific to volcanic forcing: it is not obvious that the volcanic aerosol will cool the western Pacific and Maritime Continent faster than the central or eastern Pacific or that it is actually responsible for that cooling, due to towering clouds of almost 20 km thick hovering in that region. In other words, the cooling of the Maritime Continent could still be a dynamical response. Hence, we agree with the Reviewer that to claim the existence or not of the ODT mechanism ad-hoc sensitivity simulations need to be performed.

No matter the sensitivity experiments or initial conditions, given that for the Agung eruption the ODT response does not take place (clearly for at least for the vast majority of the initial conditions), our results suggest that even if the ODT were a mechanism, it cannot be the dominant mechanism. Furthermore, the dominant mechanism may also be model dependent, as suggested by the differences between the results of Predybaylo et al 2020 and Khodri et al., 2017 in regards to the Maritime

Continent mechanism.

Nevertheless, we have also toned down our claims throughout the manuscript. The abstract (line 13 to 16 in the RM) we now reads:

*"Our results point to the volcanically induced displacement of the ITCZ as a key mechanism that drives the ENSO response, while suggesting that the other mechanisms (the ocean dynamical thermostat, the cooling of tropical northern Africa or of*

*the Maritime Continent) commonly invoked to explain the post-eruption ENSO response may be less important in our model."*

In the results section, we modified the paragraph that talk about the ODT (line 192 to 199 in the RM):

*"The most commonly invoked mechanism that explains the ENSO response after large tropical volcanic eruptions is the ODT (Clement et al., 1996) and the preferential cooling of the warm pool relative to the eastern equatorial Pacific, leading*

*to an El Niño-like response (e.g. Emile-Geay et al., 2008; Mann et al., 2005). However, in our simulations even if there is a volcanic aerosol over the equatorial Pacific and a surface cooling for all the eruptions (Figs. 1, A3, A4 and A5), we see a negative phase of the relative ENSO and an anomalous easterly wind stress developing after the Agung eruption (Figs. 2 and 3). Should the ODT mechanism be dominant in our model, we would expect the opposite response of both the relative SST and wind stress. This suggests that the ODT is likely not the leading mechanism in our model, while not ruling out that it can*

*still play a role under specific initial conditions (e.g. Predybaylo et al., 2020)."*

**In Figure R1, with all explanations, the SST anomaly does not look like La Niña. It shifts from the equator. It is strange that for the El Chichón and Pinatubo cases, the wind anomaly changes sign along the equator. One does not expect this as a result of the shift of ITCZ.**

Given that there is a superposition of the volcanic cooling and the ENSO response, when discussing the ENSO response to volcanic eruptions, one needs to look at the changes in thermocline rather than at the surface. As already shown and discussed in the previous response (Fig. R4 in previous response), the thermocline anomalies are clearly showing a La Niña-like response following the Agung eruption. Furthermore, the wind anomalies do change sign along the equator due to the ITCZ shift, that is what triggers the ENSO response. Perhaps, the reviewer is referring to the difference in DJF year 1 in the eastern Pacific with opposite anomalies between El Chichón and Pinatubo, while similar anomalies in the western Pacific. Those differences may be caused by the fact that while El Chichón aerosol is mostly confined to the NH, Pinatubo aerosol is distributed across both Hemispheres. This is likely the cause of a weaker El Niño response for the Pinatubo eruption.

We also performed a composite analysis of the wind anomalies to investigate the changing sign of the wind anomaly along the equator (Fig R2).

[Figure]

**Figure R2:** Composite analysis of the precipitation (shadings) and 10 m wind (arrows) anomaly for all the events when the ITCZ is north of its climatological location (a,c) and the events when the ITCZ is south of its climatological location (b,d) in the three reference periods. Only significant precipitation changes are shaded with an approximate 95 % confidence level using a Student *t* test. Contours show the SST anomalies following the color bar scale (solid lines for positive anomalies and dashed lines lines for negative anomalies, the 0 line is omitted). The bold line indicates the climatological location of the ITCZ.

In figure R2, we observe the same pattern of the wind anomaly that change sign along the equator in JJA and in the second year. Hence, those wind anomalies seem to be in line with the shift of the ITCZ. Moreover, the main point that we are making is that the wind anomalies are opposite in response to Agung compared to El Chichón and Pinatubo and that is why we suggest that the ITCZ shift is the driver of the response (Fig. 5). We would also like to point out that even if the pattern is less strong, the precipitation anomalies after Agung are similar to the precipitation anomaly that we see when the ITCZ is north of its climatological position and in the same way the precipitation anomalies after Pinatubo and El Chichón resemble the precipitation anomaly when the ITCZ is south of its climatological location.

**In Figure R3, how did you calculate the confidence interval? Why is it so uniform and does not depend on the season?**
For each month, the shading is twice the standard error of the mean (SE), which is obtained by calculating the standard deviation (STD) and dividing it by the square root of the number of ensembles (200). We use the SEM as we are interested in knowing the accuracy of the sample mean and whether is significantly different from 0. As the reviewer pointed out, the STD depends on the season. However, since we are using a large ensemble, when calculating the SEM, we are diving the

STD by a large number so the difference between the season become very small therefore such small variation cannot with the eyes in the figures.

Here is an example of the STD and SEM values of the Nino3.4 index for the 12 first months after Agung eruption.

| Months | June | July | Aug. | Sept. | Oct. | Nov. | Dec. | January | Feb. | March | April | May |
|--------|------|------|------|-------|------|------|------|---------|------|-------|-------|-----|
| STD | 1.08 | 1.01 | 0.98 | 0.97 | 1.04 | 1.04 | 1.18 | 1.14 | 1.05 | 0.98 | 1.00 | 1.05 |
| SEM | 0.076 | 0.072 | 0.070 | 0.069 | 0.073 | 0.078 | 0.084 | 0.081 | 0.075 | 0.0691 | 0.071 | 0.075 |

**R3. Does the model exhibit the stochastic behavior for the spring and winter eruptions? Does it show the spring**
**predictability effect?**

In this study, we are considering the total climate response to volcanic forcing with respect to the climatology as done in several studies (Khodri et al., 2017; Ohba et al., 2013; Stevenson et al., 2016). We are not separating the ENSO response into deterministic and stochastic response as our experiment set-up does not allow to do so. For doing so, experiments with volcano and no-volcano ensemble members starting from the same initial conditions as for example in Predybaylo et al.,
2020 or Pausata et al., 2015, are necessary.

While it is called "Spring Predictability Barrier", it does not mean that models have no skills in predicting the following ENSO phase. The skill of the models, of course, gets better the closer you get to the period of time you are predicting and furthermore, boreal summer is the ENSO developing season. Nevertheless, the reviewer may be aware of the footprint mechanism (Vimont et al., 2001), in which mid-latitude anomalies from the *previous winter* affect the summer tropical
atmosphere and hence ENSO. Finally, even if the eruptions occur in winter or spring the maximum aerosol load is not occurring in those months but several months afterwards and therefore still able to affect the ENSO main developing season (boreal summer). In any case, as mentioned above, here we are comparing an ensemble of 200 members with the volcanic eruption vs. the reference climate, i.e. we are comparing two climatologies and assessing whether they are significantly different.

We have modified the paragraph that discuss the presence of other in the discussion to better consider the key results that Predybaylo et al 2020 pointed out (line 254 to 272 in the RM):

*"Our work also pointed out that the ODT (Clement et al., 1996), the cooling of the Maritime Continent (Ohba et al., 2013) and of the tropical Africa (Khodri et al., 2017) are likely not the dominant drivers of the ENSO response in the MPI-GE model. Discrepancies between these previous studies and our results may occur for three reasons. First, most previous*
*studies are based on a small number of ensemble members (e.g. 5 members for 3 eruptions for the SH plume in Zuo et al. (2018)) or they heavily rely on statistical tools using a small sample of events (e.g. SEA in Liu et al. (2018)). Consequently, those results may be biased by the use of a restrained number of ensemble members. Here, our study points out the importance of a large number of ensemble members when investigating the ENSO response to volcanic eruptions. Second, different models may present different mechanisms. The fact that the above-mentioned mechanisms are not dominant is also*
*in qualitative agreement with the modeling experiments in Pausata et al. (2020), who use the Norwegian Earth System*

*Model (NorESM), suggesting that the ITCZ shift mechanism is at play in both the NorESM and the MPI-GE. Third, the role of the initial ENSO state is important for determining the ENSO response (Pausata et al., 2016; Predybaylo et al., 2017) and consequently influencing the role of the ODT and the other above-mentioned mechanisms. Our study only considers the total climate response to volcanic forcing; however, in a recent study, Predybaylo et al, (2020) separated the ENSO response to*

*volcanic eruptions into a deterministic and stochastic response. They have shown that the deterministic response is dominant for spring and summer eruptions, while stochastic response plays a major role for eruptions occurring in winter. However, our experimental design does not allow the seperation of the total climate response into the deterministic and stochastic component with further future experiments needed in which the volcano and the reference no-volcano ensemble members start from the same initial conditions."*

**L121: I'm afraid I have to disagree with the answer. The ITCZ shift affects equatorial winds. This should inevitably disturb ocean Kelvin waves.**

We agree with the Reviewer, the ITCZ shift does affect the ocean Kelvin waves via its impact on equatorial winds. What we are trying to say is that there is a volcanically induced shift of the ITCZ in the 3$^{rd}$ year after the eruption and that shift affect the intensity of the trade winds which in turn affects the ENSO state. In our previous response we were just arguing that was not the cooling of tropical Africa responsible for the Kelvin wave anomalies.

**L167: I am not sure I can completely agree with the answer. Rossby waves propagate to the west and can not affect the Pacific directly. The train of Kelvin or Rossby waves can not be excited by graduate cooling. There should be a periodic forcing, not strictly periodic but variable in time. The release of latent heat is an example of such a forcing.**

**The damping of WAM should decrease this forcing. Why should the Kelvin wave activity increase? It will change and become weaker.**

The answer that we gave to the Reviewer comes from Khodri et 2017. In that part of our manuscript, we are presenting their results and conclusions therefore we cannot alter them. We have slightly modified the text to make it clearer, and it now reads (line 75 to 80 in the RM):

*"Khodri et al., (2017) suggested that the cooling of tropical Africa, following volcanic eruptions, may increase the likelihood of an El Niño events through the weakening of the West African Monsoon and changes in the Walker circulation. More specifically, the authors suggest that the reduction of the tropical precipitation and tropospheric cooling favours anomalous atmospheric Rossby and Kelvin waves in autumn (SON), with a weakening of the trade winds over the western Pacific, leading to El Niño-like conditions in the year after the eruption."*

**Response to Reviewer #2:**

**The paper analyses one of the biggest datasets - the MPI grand ensemble - in terms of ENSO responses to strong volcanic forcing peaking at different latitudes. The topic is hot and challenging, the chosen dataset is comprehensive and budding; however, the results require further investigation.**

We thank the reviewer for the thorough evaluation of the manuscript and appreciate the positive remarks on the pertinence of the study.

**Major comments:**

**1. The paper reports warming of the tropical Pacific as a response to both symmetric and Northern Hemisphere**
**volcanic forcings, and cooling of the tropical Pacific after Southern Hemisphere volcanic forcing. Despite being statistically significant, the obtained amplitudes of warming (0.3 degrees) and cooling (-0.2 degrees) anomalies are too small to be interpreted as El Niño-like and La Niña-like responses. El Niño and La Niña occur when the sea surface temperature anomaly is above 0.5 degrees for a few consecutive months. Moreover, the anomalies are calculated as the relative Niño3.4 index, not the conventional Niño3.4 index. This index is used to detect the "dynamic El Niño**
**response" by removing the mean tropical cooling. Naming this result "relative El Niño-like response" may confuse the reader because the range -0.5 to 0.5 degree is, in fact, a neutral state of the ocean.**

First, we would like to clarify that for Niño3.4 index the threshold is +/-0.4°C Nino SST Indices (https://climatedataguide.ucar.edu/climate-data/nino-sst-indices-nino-12-3-34-4-oni-and-tni). Second, there is no such convention for El Niño-like as there is for El Niño and Niña. The **"-like"** indicates that **warm or cold anomalies in the**
**equatorial Pacific** will be superimposed to the ongoing ENSO state. Furthermore, as we mentioned in the first round of reviews, the -like is to indicate that the anomalies resemble the El Niño anomalies in terms of structure and pattern. It has been standard practice to name it this way (e.g., Pausata et al., 2015, 2016, 2020 and a recent review paper by McGregor et al., 2020) and we have specified it in the text. Furthermore, even without using the RSST the magnitude of the anomalies are similar to CMIP (Maher et al., 2015) (Fig. A13).

To clarify the point, we now clearly mention in the beginning of the manuscript the meaning of the "relative La Niña-like" and "relative El Niño-like" (line 45 to 47 in the RM):

*"In this study, the terms El Niño-like and La Niña-like conditions are used to describe an anomalous warming or cooling of the equatorial Pacific relative to a climatology and the term relative is used to describe deviations from the tropical average (20°N–20°S)."*

**2. The authors discuss only the sensitivity of the ENSO response to the spatial distribution of the volcanic aerosols. They analyze the ensemble mean anomaly and disregard the important sensitivity to the initial conditions and timing**

**of the eruptions. Predybaylo et. al (2020) conducted a study with 54 100-member ensembles specifically analyzing the ENSO sensitivity to initial conditions, timing, and strength of the tropical eruptions with the symmetric aerosol distribution. It showed that disregarding the initial conditions and timing of the eruption in the analysis may lead to**
**improper interpretation of the data, related mechanisms and therefore confusing results. Thus, only the ensemble size cannot be a decisive factor in the conducted analysis.**

We agree with the reviewer on the importance of the initial conditions. As per our answer to Reviewer #1, since we do not test impacts of the initial conditions or the timing of eruption as this is out of scope of the study, we further toned our claims in the manuscript (see also answer R1 and R3 to Reviewer #1).

We would like to point out that sampling the initial conditions is more complicated than just looking at the SST when the eruption occurs. To truly sample what the model would have done without the volcano more experiments need to be done (e.g., volcano and no-volcano starting from the same initial conditions). Given that Predybaylo et al 2020 argue that the stochastic response is dependent on the initial condition and the eruption time, it really is out of the scope of what we can do with the experiments. The objective of this paper is indeed to investigate the sensitivity of the ENSO response to the spatial
distribution of the volcanic aerosols.

**3. The ODT mechanism depends on the initial conditions (Predybaylo et al. 2020) and therefore cannot be easily rejected in this paper. The other mechanisms also require further investigation. For example, the eruption of Agung happened in winter and its impact may be too noisy for interpreting the mechanism and therefore the ITCZ shift in**
**this case could be questioned. Additionally, the amplitude of the response is again too small for being La Niña-like.**

We agree with the reviewer that the ODT *mechanism* cannot be completely rule out based on our results and it may exist and play a role under specific initial conditions. No matter what, given that for the Agung eruption the ODT *response* does not take place, our results suggest that even if the ODT were a mechanism, it cannot be the dominant mechanism. This is a key results and we suggest that following Predybaylo's studies more investigation needs to be done looking at the initial
conditions. See also answer R3 to Reviewer #1.

Regarding the time of the eruption for the Agung: the effect of the eruption and its associated aerosol loading is not limited to the time of the eruption. As it is possible to see from the AOD figure (Fig.1), the aerosol loading peaks in summer and high aerosol concentrations persist for a couple of years and so does the cooling. Furthermore, the anomalies are significant, which means it is unlikely to be noise.

Regarding the amplitude of the response, we have clarified it above (see major comment 1 to reviewer #2).

**Minor comments:**

**L48: abbreviations should be spelled out**

We thank the reviewer for pointing that out.

**L49: the abbreviation should read: CESM-LME**

We thank the reviewer for pointing that out.

**L52: the long name of CESM-LME should be removed as it was already defined**

We thank the reviewer for pointing that out.

**L55: reference case - climatology or control (experiments without volcanic forcing)?**

Here we changed the term reference case to climatology.

**L50-51: Liu Li reference should read Liu et al.**

We thank the reviewer for pointing that out.

**L59: The mechanism is not quite clearly described, could you please rephrase it. Also it should be mentioned that this mechanism depends on the ocean preconditioning (Predybaylo et al. 2017, 2020)**

Here we clarified our explanation about the ODT mechanism (line 57 to 61 in the RM):

*"One of the most frequently used hypotheses is the "ocean dynamical thermostat" mechanism (ODT)* (Clement et al., 1996), *where a preferential cooling in the western Pacific relative to the eastern Pacific takes place. Such a differential cooling weakens the zonal SST gradient along the equatorial Pacific which causes a relaxation of the trade winds, leading to a temporary weakening of the ocean upwelling in the eastern Pacific. This process is then amplified by the Bjerknes feedback, yielding an El Niño* (Bjerknes, 1969).*"*

About Predybaylo et al. 2020, we now included a specific paragraph in the introduction that talk about this paper (line 93 to 97 in the RM):

*"A recent study by Predybaylo et al, 2020 additionally demonstrates the need for using large ensembles to investigate the ENSO response to volcanoes, utilising 54 100-member ensembles of idealised experiments, they found that the role of the initial ENSO state as well as decomposing the response into a deterministic and stochastic component are important specially to highlight the role of the ODT mechanism"*

**L57: weakens or amplifies**

We thank the reviewer for pointing that out

**L74: shifts away**

We thank the reviewer for pointing that out

**L76: larger cooling compared to what?**

We added compared to the NH in the sentence, we now read :

*"In contrast, the ITCZ moves northward following a larger SH cooling compared to NH, strengthening the trade winds and triggering La Niña-like anomalies"*

**L88: This is not true. Predybaylo et al. 2020 analyses 54 100-member ensembles.**

As mentioned in a previous comment, now dedicate a specific part of the introduction on the Predybaylo et al 2020 paper.

(See minor comment L59 of the second reviewer).

**L86-87: The stochastic component of the ENSO response strongly depends on the timing of the eruption.**

Like mentioned in the previous comment, we have now highlighted the importance of the stochastic component referring to Predybaylo et al 2020 paper (See minor comment L59 of the second reviewer).

**L95: The large ensemble of mixed initial conditions cannot fully explain the mechanisms and ODT, for example, depends on the initial conditions.**

We thank the reviewer for bringing that out, as we mentioned before (see also answer R1 and R3 to Reviewer #1), we toned down our conclusion on the ODT because we do not test the initial conditions appropriately.

**L112: not all the studies use composites, for example Ohba et al. 2013, Predybaylo et al. 2017, 2020**

We thank the reviewer for pointing that out, that was indeed not what we meant. We modified the sentence, we now read :

*"Such a large ensemble also allows us to analyse the ENSO response of individual eruptions instead of a composite of multiple eruptions as done in some previous studies that used a small number of ensembles (e.g. Maher et al., 2015;*

*Stevenson et al., 2016; Zuo et al., 2018)."*

**L122: By default, the RSST enhances the ENSO response and should only be analysed together with the SST (Predybaylo et al. 2020).**

We had a similar comment by a previous reviewer and we now include our analysis with the SST in the Appendix. To clarify that point, we added to the methodology:

*"In this study, we use the RSST to isolate the intrinsic ENSO signal, but the analysis using the SST are available in the Appendix (Khodri et al., 2017)."*

[revised manuscript text omitted]

---

## Author Response (AR3)

**Response to the Editor**

We would like to thank the editor for the time and effort taken to provide this review of our manuscript. We have addressed all the minor comments raised by the editor. In doing so, we feel we have crafted a better revised manuscript.

Below, we copy the editor's comments in bold and describe how each of these issues has been addressed. The revised manuscript is attached after the answers to the editor and the changes compared to the previous version are highlighted in bold.

**Response to the Editor:**

**General Comments :**

**I'd like to see more comparison to the results of the CESM-LME. Do your conclusions agree or disagree? That is, how**
**model-dependent are your answers?**

Our conclusions overall agree with Stevenson et al. (2016) that uses all ensemble members of CESM-LME (14) that show a La Nina-like response in the first winter following SH eruption and an El Nino-like response following tropical (second winter) and NH (first and second winter) eruptions. However, they disagree with the studies of Zuo et al. 2018 and Liu et al. (2018) who used, respectively, a subset of the total CESM-LME (5) and an earlier version of the model (CESM1.0 vs CESM1.1) with
only 1 ensemble member.

To clarify how our results are model dependent, we added a comparison to recent studies that use the CESM-LME to the discussion. We now read in the manuscript:

*"Discrepancies between these previous studies and our results may occur for three reasons. First, different models may present*
*different mechanisms. The fact that the above-mentioned mechanisms are not dominant is also in qualitative agreement with the modeling experiments in Pausata et al. (2020), who use the Norwegian Earth System Model (NorESM). Another study using all the ensemble members (14) of a large ensemble (CESM-LME) is also in overall agreement with our results (Stevenson et al. 2016), finding a La Nina-like response to SH eruption and an El Nino-like response to NH and tropical eruption. However, other studies using either a subset (5 members) of the CESM total ensemble (Zuo et al. 2018) or an early version of*
*CESM with only one ensemble (Liu et al., 2018) display El-Nino like anomalies for all type of eruptions, which brings us to the following point. Second, most previous studies are based on a small number of ensemble members (e. g. 5 members for 3 eruptions for the SH plume in Zuo et al. (2018)) or they heavily rely on statistical tools using a small sample of events (e. g. SEA in Liu et al. (2018)). Consequently, those results may be biased by the use of a restrained number of ensemble members. Here, our study points out the importance of a large number of ensemble members when investigating the ENSO response to*
*volcanic eruptions."*

**Not long after the eruption of Pinatubo, there was the eruption of Cerro Hudson in the southern hemisphere. It was small and probably didn't affect your results, but you should say something about it, especially because your study is about hemispheric differences in eruptions.**

The aerosol data set comes from Stenchikov et al 1998, where satellite measurements from SAGE-II are used. The satellite does not separate the aerosol from Pinatubo of the aerosols from the Cerro Hudson eruption, therefore, the Pinatubo aerosol distribution used in our study also includes the Cerro Hudson aerosol plume. The superposition of the two eruptions still leads to a symmetrical aerosol distribution and that is what is prescribed in the model (Fig. 1). We now mention in the methodology that the aerosol data set comes from satellite observations to clarify that point.

*"The stratospheric aerosols used in this study are prescribed in the historical simulations of the model (MPI ESM) from an extended version of data set of Stenchikov et al. (1998) (Giorgetta et al., 2013). The data set consists in monthly averages of the radiative properties of the aerosols such as the single scattering albedo, the aerosols extinction and the asymmetry factor and they are based on satellite observation of Pinatubo's eruption (Schmidt et al., 2013)."*

*"The reasons why similar eruptions can lead to different aerosol distributions are still being investigated, but some causes are the location of the volcanoes, the season and the strength of the eruption (Stoffel et al., 2015; Toohey et al., 2011), the Quasi-Biennial Oscillation and local meteorology (e.g. Choi et al. 1998), which determine wind direction and the aerosols distribution across the equator. Regarding the Pinatubo aerosol distribution, we note that the eruption was followed few*

*months after by the eruption of Cerro Hudson (Chile) in the Southern Hemisphere. As the aerosol distribution for Pinatubo is based on satellite observations, the aerosol properties prescribed to the model also include Cerro Hudson eruption"*

**Minor Comments:**

**Typo on lines 34 and 134**

We thank the editor for pointing that out.

**Lines 75 and 94: use cite instead of citep**

The citations have been corrected.

**I think Section 2 needs more clarification. More specifically, it has been shown that models need a sufficiently high top and good vertical resolution are necessary to properly represent many of the surface climate effects of volcanic eruptions (e.g., Driscoll et al., 2012). A model with only 16 vertical layers is worrisome (although you do have the winter warming that is expected), and there is no description of how the volcanic aerosols are represented in the model.**

We agree with the editor that only 16 levels may seem worrisome. However, as shown by Suarez-Gutierrez et al., 2021, MPI-

GE has been found to be the best model within the 10 analysed in their study in representing both global and regional internal variability and forced response. Therefore, the limited number of vertical levels does not translate into a poor representation of the forced response associated with volcanic eruptions. We have included the following paragraph in Section 2:

*"... (e.g. Robock & Mao, 1995; Timmreck, 2012). Furthermore, while it has been argued that models need a sufficiently high*
*top and good vertical resolution are necessary to properly represent many of the surface climate effects of volcanic eruptions (e.g., Driscoll et al., 2012),(Suarez-Gutierrez et al., 2021) analysed 10 different global climate model and found that MPI-GE was the best model in representing both global and regional internal variability and forced response. Therefore, the limited number of vertical levels does not translate into a poor representation of the forced response associated to volcanic eruptions."*

Moreover, like mentioned in our previous comment we added some details about how the volcanic aerosols are represented in the model:

*"The stratospheric aerosols used in this study are prescribed in the historical simulations of the model (MPI ESM) from an extended version of data set of Stenchikov et al. (1998) (Giorgetta et al., 2013). The data set consists in monthly averages the*
*radiative properties of the aerosols such as the single scattering albedo, the aerosols extinction and the asymmetry factor and it is based on satellite observation of Pinatubo's eruption (Schmidt et al., 2013)."*

**Lines 139-141: And also things like the QBO and local meteorology, which determine wind direction and how far the aerosols spread or whether they spread across the equator**
The QBO is indeed very important in determining the spread and distribution of the aerosol following volcanic eruptions. We have modified the sentence to read:

*"The reasons why similar eruptions can lead to different aerosol distributions are still being investigated, but some causes are the location of the volcanoes, the season and the strength of the eruption (Stoffel et al., 2015; Toohey et al., 2011), the*
*Quasi-Biennial Oscillation and local meteorology (e.g. Choi et al., 1998), which determine wind direction and the aerosols distribution across the equator."*

**Lines 170-171: Can you do better than "could"? Your ensemble has a high enough signal-to-noise ratio that you could probably do a good job with figuring this out.**
We thank the editor for his comment. Previous reviews asked us to tone down our claims but we now removed the "could" to have a stronger statement as the ensemble of our results together with previous studies support such conclusion.

*"This indicates that the ITCZ response is sensitive to the magnitude and spatial distribution of the forcing."*

**100** **Lines 198-199: I agree that it appears as though there are other things going on in your model than ODT. My next question – is that right? Do you have a way of validating this? Same with the cooling of the Maritime Continent. If your model shows results that are counter to long-established literature, it's either because you've discovered something new or that your model doesn't do a good job with this. It would be prudent to convince readers that it's not the latter. This will require showing more than the fact that your model reproduces cooling patterns.**

**105** The hypothesis that the ODT mechanism may not be a mechanism at play in the ENSO response following volcanic eruption has been brought forward already in several studies that seriously that questioned the existence of this mechanism (e.g. Pausata et al. 2020, Khodri et al. 2017). The Maritime Continent (MC) hypothesis was proposed in (Ohba et al., 2013), however, in their Figure 4 is not possible to see the larger cooling of the MC relative to the surrounding ocean justifying the wind anomalies. Furthermore, to our knowledge, Khodri et al., 2017 is the only study that tried to test it could not support this theory. Finally,

**110** there are also several studies that use different models that also pointed out the role of the ITCZ-shift mechanism (Colose et al., 2016; Pausata et al., 2015, 2016, 2020; Stevenson et al., 2016). Therefore, we do not think that we are countering long-established literature, our results, using a different model, are more supporting the most recent studies on the subject. Finally, our conclusions are not only based on the temperature anomaly pattern, but also on the wind, precipitation and sea level pressure that match the expected ITCZ shift mechanism.

**115** Only through a set of specific sensitivity simulations would be possible to accurately test the ODT and MC mechanism. However, such an experiment in a large ensemble set-up would be extremely demanding and we do not have the computing resources to perform them.

**Lines 224-231: I'll admit that I found this paragraph unsatisfying. It essentially reports simulation results and then**
**120** **uses vague words like "suggesting" and "could". You also use "point to" in line 234. And early in this paragraph, you note a mismatch between your mechanism and SLP anomalies. I agree that doing a good job with this would require sensitivity experiments. But this is kind of your main argument, and I was hoping you'd do a better job of convincing me that your argument is right, rather than just a hypothesis. Perhaps I've misunderstood your point, and this is simply a matter of rephrasing.**

**125** All the analysis that we have performed supports our main conclusion, i.e. the ITCZ shift mechanism. However, we have toned down our phrasing as per request of the previous reviewers, that's why we use *"strongly point to the volcanically-induced ITCZ displacement as the primary driver of the ENSO response following volcanic eruptions"*.

In paragraph 224-231, we are bringing forward an hypothesis on why we see different results compared to Pausata et al. (2020) with regards to the extratropical-to-tropical teleconnection, as it is a speculation on why the two models behave differently we
**130** should be cautious in the language we use. However, this is not our main argument, but rather a side, additional mechanism recently proposed by. The main argument and mechanism is the ITCZ shift that we supported with several analyses.

**References**

[revised manuscript text omitted]